

# Reconstructions of the 1900–2015 Greenland ice sheet surface mass balance using the regional climate MAR model

Xavier Fettweis[1], Jason E. Box[2], Cécile Agosta[1], Charles Amory[1], Christoph Kittel[1], and Hubert Gallée[3]

[1]Laboratory of Climatology, Department of Geography, University of Liège, Liège, Belgium
[2]Department of Glaciology and Climate, Geological Survey of Denmark and Greenland (GEUS), Copenhagen, Denmark
[3]Laboratoire de Glaciologie et Géophysique de l'Environnement (LGGE), Grenoble, France

*Correspondence to:* Xavier Fettweis (xavier.fettweis@ulg.ac.be)

**Abstract.**

With the aim of studying the recent Greenland Ice Sheet (GrIS) Surface Mass Balance (SMB) decrease with respect to the last century, we have forced the regional climate MAR model (version 3.5.2) with the ERA-Interim (1979–2015), ERA-40 (1958–2001), NCEP-NCARv1 (1948–2015), NCEP-NCARv2 (1979–2015), JRA-55 (1958–2014), 20CRv2(c) (1900–2014) and ERA-20C (1900–2010) reanalysis. While all of these forcing products are reanalyses assumed to represent the same climate, they produce significant differences in the MAR simulated SMB over their common period. A temperature adjustment of $+1^oC$ (respectively $-1^oC$) improved the accuracy of MAR boundary conditions from both ERA-20C and 20CRv2 reanalyses given that ERA-20C (resp. 20CRv2) is $\sim 1^oC$ colder (resp. warmer) over Greenland than ERA-Interim over 1980–2010. Comparisons with daily PROMICE near-surface observations validated these adjustments. Comparisons with SMB measurements from PROMICE, ice cores and satellite derived melt extent reveal the most accurate forcing data sets for simulating the GrIS SMB to be ERA-Interim and NCEP-NCARv1. However, some biases remain in MAR suggesting that some improvements need still to be done in its cloudiness and radiative scheme as well as in the representation of the bare ice albedo.

Results from all forcing simulations indicate: i) the period 1961–1990 commonly chosen as a stable reference period for Greenland SMB and ice dynamics is actually a period when the SMB was anomalously positive ($\sim +10\%$) compared to the last 120 years; ii) SMB has decreased significantly after this reference period due to increasing and unprecedented melt reaching the highest rates in the 120 year common period; iii) before 1960, both ERA-20C and 20CRv2 forced MAR simulations suggest a significant precipitation increase over 1900–1950 although this increase could be the result of an artefact in reanalysis not enough constrained by observations during this period. These MAR-based SMB and accumulation reconstructions are however quite similar to those from Box (2013) after 1930, which confirms the Box (2013)'s stationarity assumption of SMB over the last century. Finally, the ERA-20C forced simulation only suggests that SMB during the 1920–1930 warm period over Greenland was comparable to the SMB of the 2000's due to both higher melt and lower precipitation than normal.



## 1 Introduction

Since the end of the 1990's, the Greenland Ice Sheet (GrIS) has been losing mass as a result of both surface meltwater run-off increase (representing ∼60% of the mass loss) and iceberg discharge increase (van den Broeke et al., 2016). This recent acceleration of the ice dynamics is likely a consequence of the increase of meltwater availability and ocean warming although

the role of meltwater remains unclear (Sundal et al., 2011; Tedstone et al., 2013). The recent surface melt increase likely results from global warming, enhanced by the Arctic amplification (Serreze et al., 2011) and general circulation changes in summer favouring advection of warm air masses over GrIS (Fettweis et al., 2013a; Hanna et al., 2016). In view of the impact of GrIS on the observed global sea level rise (van den Broeke et al., 2016), it is important to consider this recent surface mass loss in a longer perspective.

The warming observed in the 1930's (Chylek et al., 2006) is for example often considered as equivalent of the warming observed since the end of the 1990's suggesting that the recent surface mass loss could not be unprecedented in the 100-yr Greenland climate history. A first estimation of the Surface Mass Balance (SMB) through the whole previous century has been made by Fettweis et al. (2008), Hanna et al. (2011) and more recently by Box (2013b) based mainly on observations and statistical regressions and corrections. However, the amount of in situ observations are too sparse over Greenland before

1950 to build reconstruction at the daily time scale as in Mernild and Liston (2013) and many uncertainties remain in these reconstructions available at the monthly time scale at best. The recent development of new reanalysis data set covering the whole last century and constrained by observations from both Greenland and outside Greenland offers a new opportunity to evaluate SMB over GrIS through the last century (Hanna et al., 2011). Regional climate models (RCMs) especially developed for polar regions (Rae et al., 2012) and coupled with complex snow models are attractive tools to physically downscale the

6-hourly reanalyses and estimate SMB. Both the spatial resolution and the snow model used in reanalyses are not yet adequate to directly derive SMB from reanalyses (Cullather et al., 2016). Among the "polar" RCMs, there is the model MAR (for Modèle Atmosphérique Régional in French) fully coupled with a snow energy balance model, developed and extensively validated to study the present Greenland climate as well as to perform future projections of GrIS SMB for the last IPCC report (Fettweis et al., 2013b).

In the present manuscript, we use the model MAR forced by 8 reanalyses over Greenland at a resolution of 20km for the period 1900–2015: i) to evaluate the uncertainties coming from reanalyses in RCM-based reconstructions (while reanalyses are assumed to represent exactly the same climate) and ii) to estimate SMB before 1950 with two new reanalyses covering the whole century. All RCM's based SMB estimations use the ECMWF (European Center for Medium range Weather Forecasting) reanalysis as forcing until now and cover only the second half of the last century (e.g. Lucas-Picher et al. (2012); Rae et al.

(2012); Noël et al. (2016)).

After a brief description of the MAR model and reanalyses used as forcing in Sect. 2, Sect. 3 compares the 8 reanalyses used for MAR forcing over Greenland as well as the reanalysis forced MAR results over our reference period (1980–1999). In Sect. 4, we present a validation over 1958–2010 of the 8 MAR–based series versus in–situ observations from ablation stakes,





ice cores and satellite derived melt extent. Finally, Sect. 5 discusses the time evolution of the GrIS SMB since 1900 as well as the uncertainties coming from the reanalyses.

## 2 data

### 2.1 The MAR model

The version of MAR used here is 3.5.2 (called MAR hereafter) and has been used by Colgan et al. (2015); Alexander et al. (2016); Koenig et al. (2016); Schlegel et al. (2016); Wyard et al. (2016). See Fettweis (2007), Reijmer et al. (2012) and Fettweis et al. (2013b) for a detailed description of the MAR model and its surface scheme SISVAT (Soil Ice Snow Vegetation Atmosphere Transfer) dealing with the energy and mass exchanges between surface, snow and atmosphere. With respect to the version 2 of MAR (called MARv2 hereafter) and set-up used in Fettweis et al. (2013b),

- A resolution of 20 km (instead of 25 km) as well as the GrIS topography from Bamber et al. (2013) (instead of Bamber et al. (2001)) were used here. In addition, to each atmospheric MAR $20 \times 20$ km$^2$ grid cell, we associated two sub-grid cells covered by tundra and permanent ice according to the Bamber et al. (2013) ice mask. This factional ice sheet mask in MARv3.x allows to compute SMB outside the original MAR ice sheet mask (in the aim of forcing ice sheet models at higher resolutions) and a grid cell will be considered hereafter as an ice sheet grid cell if its permanent ice cover is higher than 50 %. In addition, when integrated over the whole ice sheet, the surface mass values will be weighted by the permanent ice cover of each grid cell (covered at least by 50 % of permanent ice).

- According to the MAR bare ice albedo overestimations found by Alexander et al. (2014) using MARv3.2, the bare ice albedo has been improved in MARv3.5.2 by exponentially varying between 0.4 (dirty ice) and 0.55 (clean ice) as a function of the accumulated surface water height and slope. For density lower than 550 kg/m$^3$, the CROCUS albedo (Brun et al., 1992) is used with a minimum albedo value sets to 0.7. Concerning snowpack with surface density higher than 550 kg/m$^3$ (representing the maximum density of pure snow), the minimum allowed albedo is a linear function making a smooth transition between the minimum pure snow albedo (0.7) and clean ice albedo (0.55).

- Vernon et al. (2013) emphasized an overestimation of accumulation simulated by MARv2 in the interior of the ice sheet. This bias was in part corrected in MARv3.5.2 by slightly increasing the snowfall velocity allowing to have more precipitation along the ice sheet margin and less inland.

- Finally, MARv3.5.2 is now parallelized with OpenMP, its outputs are CORDEX compliant in addition to the usual bug corrections since MARv2.



## 2.2 Reanalyses

In this study, in order to force MAR every 6 hours at its lateral atmospheric boundaries (temperature, humidity, wind and pressure at each vertical MAR level) and over oceanic grid cells (sea ice cover and sea surface temperature), we use the reanalyses listed below:

- ERA-Interim over 1979–2015 available at a resolution of ∼0.75° degree from ECMWF. As in Fettweis et al. (2013b), this state-of-the-art 3rd generation reanalysis is used as reference over our chosen reference period (1980–1999) and assimilates all of the in-situ and remote observations available (Dee et al., 2011).

- ERA-40 over 1958–2001 (resolution: ∼1.125° degree), the 2nd generation reanalysis from ECMWF (Uppala et al., 2005). One of the main improvements made with ERA-Interim with respect to ERA-40 is a fully revised humidity scheme (Dee et al., 2011) which impacts the snowfall amount simulated by MAR as shown by Fettweis et al. (2013b).

- ERA-20C over 1900–2010 (resolution: ∼1.25° degree), the latest generation of ECMWF reanalysis products assimilating only surface pressures as well as surface marine winds but starting in 1900 (Poli et al., 2016). As this reanalysis assimilates much less data than ERA-40/ERA-Interim, it is generally less reliable than the other ECMWF reanalyses but its reliability increases with time with the increasing amount of assimilated observations.

- NCEP-NCARv1 (referred as NCEPv1 here) over 1948–2015 (resolution: 2.5° degree), 1st generation reanalysis from the NCEP-NCAR covering the half of the last century at low spatial resolution (Kalnay et al., 1996).

- NCEP-DOE (referred NCEPv2 here) over 1979–2015 (resolution: 2.5° degree), 2nd generation reanalysis using an improved version of the NCEP-NCARv1 global model and assimilating in addition satellite data with respect to NCEP-NCARv1 (Kanamitsu et al., 2002).

- 20CRv2 over 1871–2012 (resolution: 2.0° degree), experimental reanalysis based on a ensemble mean of 56 members assimilating only surface pressure, monthly Sea Surface Temperature (SST) and Sea Ice Cover (SIC) (Compo et al., 2011). Only outputs after 1900 were used here. As for ERA-20C, its reliability increases with time (i.e. with the number of assimilated data).

- 20CRv2c over 1851–2014 (resolution: 2.0° degree), same as 20CRv2 but correcting a bias found in the sea ice distribution by assimilating new SST and SIC data.

- JRA-55 over 1958–2014 (resolution: 1.25° degree), 2nd generation reanalysis from the Japan Meteorological Agency (JMA) described in Kobayashi et al. (2015).



## 3 Evaluation over 1980-1999

### 3.1 MAR forcings

In Fettweis et al. (2013b), summer temperatures at 600 hPa, geopotential height at 500 hPa, wind speed at 500 hPa from the different MAR forcing fields were compared over 1980–1999 to explain the discrepancies between the MAR simulations using different forcings.

According to Fettweis et al. (2013a), the variability of the JJA (June-July-August) mean 700 hPa temperatures (T700) in the free atmosphere over Greenland drives the melt variability in MAR. Therefore, temperature biases at the MAR boundaries will directly impact the amount of melt simulated by MAR (Fettweis et al., 2013b). Since free atmosphere temperatures are not assimilated in both 20CRv2(c) and ERA-20C reanalysis, a comparison of this field is presented in Fig. 1 over our reference period (1980–1999) covered by all data sets used here and through that one SMB has been stable. As justified in Fettweis et al. (2013b), a comparison over a longer period than 20 years does not change the conclusions of this comparison.

The mean 1980–1999 free atmosphere JJA temperature from ERA-40 and NCEPv1 compares very well with ERA-Interim over Greenland (see Figs. 1b and 1e). Surprisingly, the comparison is worse with the 2nd generation of the NCEP reanalysis which is warmer than ERA-Interim in summer except at the South-East of Greenland (see Fig. 1f). As specific humidity (used as forcing at the MAR lateral boundaries) needs to be derived from relative humidity in NCEPv2, these temperature biases impact the precipitation amount simulated by MAR forced by NCEPv2. The reanalyses covering the whole 20th century (i.e. 20CRv2 and respectively ERA-20C) are significantly ( $> 1°C$ ) too warm (see Fig. 1c) and too cold (Fig. 1g) in summer respectively by comparison with ERA-Interim. Similar anomalies also occur in winter (not shown here). In view of these biases in 20CRv2 and ERA-20C, a correction of $-1°C$ (resp. $+1°C$ ) was applied to the temperature fields from these two reanalyses at each vertical level of the MAR lateral boundaries by keeping the relative humidity constant. These corrected reanalyses will be called 20CRv2-corr and ERA-20C-corr hereafter. The aim of these corrections is to have a good agreement with the ERA-Interim forced MAR melt rate over the last decades to be able to compare the recent melt increase with past conditions. As the melt response to a temperature anomaly is not linear (Fettweis et al., 2013b), inaccurate current melt rates bias melt anomalies in the past. It should be noted that no change was applied to the MAR oceanic boundaries (SST and SIC) and that the temperatures corrections were homogeneously applied through the whole year and over the whole period covered by these two reanalyses as these biases are constant in time over 1948–2010 with respect to NCEPv1 (not shown here). As we will see in Section 4.1, these temperature corrections enable a better comparison of MAR with in situ temperature measurements using unmodified 20CRv2 and ERA-20C based fields as lateral boundaries. Finally, as the warm bias from 20CRv2 is partly corrected in 20CRv2c and is now centred around zero with a too warm atmosphere at the north of Greenland and too cold at the south-west, unmodified 20CRv2c temperatures are then used to force MAR at its lateral boundaries.

Since the surface pressure has been assimilated in all reanalyses used here, the general circulation (gauged here by the 500hPa geopotential height in Fig. 2) including the North Atlantic Oscillation (NAO) compares well over the recent decades (Belleflamme et al., 2013) except for 20CRv2(c) which underestimates wind speed at 500 hPa inducing anti-clockwise circulation anomalies over Greenland (see Figs. 2g and 2h). Moreover, as a consequence of the lack of (or less reliable) assimilated





data before 1940, the general circulation variability from 20CRv2 and ERA-20C diverges according to Belleflamme et al. (2013) and explains the discrepancies between MAR forced by 20CRv2(c) and ERA-20C before 1940 (See Section 6).

## 3.2 MAR results

With respect to the interannual variability, the mean SMB components from the different MAR simulations compare very well when they are integrated over the whole ice sheet except for the non-corrected ERA-20C and 20CRv2 forced MAR simulations (see Table 1). However, when looking to spatial differences (see Figs. 3 and 4), the comparison with the $MAR_{ERA-Interim}$ simulation over 1980–1999 shows that:

1. $MAR_{ERA-40}$ overestimates slightly precipitation because the ERA-40 high atmosphere is wetter than ERA-Interim as a result of biases found in the ERA-40 humidity scheme and corrected afterwards in ERA-Interim (Dee et al., 2011). However, this wet anomaly is homogeneous over the whole integration domain explaining why there are not locally significant discrepancies between $MAR_{ERA-Interim}$ and $MAR_{ERA-40}$.

2. Although the temperature corrections of $+1^o$ degree at the MAR lateral boundaries reduce the underestimation of melt by $MAR_{ERA-20c}$, $MAR_{ERA-20c-corr}$ is still too cold in summer. Both ERA-20C forced simulations also significantly underestimate precipitation along the south-western coast because no enough humidity is advected at the south-west lateral boundaries of our integration domain, from where the prevailing flow over South-Greenland comes. This too dry and cold main flux is a consequence of the ERA-20 underestimation of the free atmosphere temperature and wind speed in this area (see Figs. 1c and 2c).

3. Most of the differences between $MAR_{ERA-Interim}$ and $MAR_{NCEPv1}$ (resp. $MAR_{JRA-55}$) are within the interannual variability of $MAR_{ERA-Interim}$ over 1980–1999 and are therefore insignificant. However, both simulations underestimate precipitation along the south-east coast with respect to $MAR_{ERA-Interim}$.

4. $MAR_{NCEPv2}$ is too wet (resp. too dry) at the south-west (resp. south-east) of the ice sheet although the general circulation (here Z500) from NCEPv2 compares very well with ERA-Interim. However, NCEPv2 is too warm (resp. too cold) at the south-west (resp. south-east) of Greenland which impacts the amount of humidity advected by MAR from its lateral boundaries. This is because the specific humidity is derived from the NCEPv2 relative humidity and then, affected by the temperature biases found in NCEPv2 with respect to ERA-Interim.

5. The pattern of anomalies from $MAR_{20CRv2-corr}$ and $MAR_{20CRv2c}$ with respect to $MAR_{ERA-Interim}$ are the same and mainly results from anomalies in precipitation. We can see that the temperature correction in 20CRv2-corr as forcing reduces the $MAR_{20CRv2}$ run-off overestimation ($+40\%$) versus $MAR_{20CRv2-corr}$ but this correction does not impact the simulated MAR precipitation: $MAR_{20CRv2(c)}$ is too wet (resp. dry) along the north-eastern (resp. north-western) coast as a result of the anti-clockwise circulation anomalies simulated by 20CRv2(c) with respect to ERA-Interim (see Fig. 2). Finally, except along the south-western margin where 20CRv2-corr and 20CRv2c are too cold in summer, $MAR_{20CRv2-corr}$ and $MAR_{20CRv2c}$ weakly overestimates run-off with respect to $MAR_{ERA-Interim}$.



## 4 Validation with the PROMICE data sets

### 4.1 Near-surface climate

As validation of the near-surface conditions simulated by MAR, a comparison with daily measurements from the Automatic Weather Station (AWS) of the network PROMICE (Programme for Monitoring of the Greenland Ice Sheet) (Ahlstrom et al., 2008) is presented over the common period covered by the forcing data sets used here: 2008–2010. The raw PROMICE data are used here without any filtering or withdrawing of aberrant values. The MAR values at each station are based on an interpolation of the 4 nearest MAR grid cells weighted by the inverse distance to the station. As the elevation difference between MAR and AWS is not corrected, the comparison is only carried out on the 12 AWS's listed in Table 2 with an elevation difference less than 100m with the interpolated MAR 20km topography. Scatter plots are shown in Fig. 5 and statistics are listed in Table 3.

In average over the 12 AWS's, the comparison of $MAR_{ERA-Interim}$ with the measured daily near-surface temperature is excellent with a correlation above 0.96 and a RMSE (Root Mean Square Error) of 2-3°C, representing less than 30% of the daily variability. The improvements with respect to MARv2 are evident. The biases with the downward short (resp. long)-wave radiation remain however high in both MAR versions with a RMSE representing 25% (resp. 70%) of the daily variability of these fluxes. Due to an underestimation of the cloudiness, MAR overestimates slightly (resp. highly underestimates) downward short (resp. long) wave radiations. Such biases in the short/long-wave were also found in the regional RACMO2.3 model (Van Tricht et al., 2016) suggesting that improvements are still needed in the clouds and/or radiative schemes of the (regional) climate models. As a result, $MAR_{ERA-Interim}$ is slightly too cold (−0.29°C at the annual scale), in particular in summer (−0.65°C) when the underestimation of the downward infrared flux is the highest (bias of −18W/m$^2$ knowing that the daily variability is 43W/m$^2$). Finally, MAR overestimates the bare ice albedo as it is limited to 0.40 in MARv3.5.2 while values ranging from 0.2 to 0.4 (due to the presence of impurities not taken into account into MAR) are observed in some PROMICE AWS's (Tedesco et al., 2016). In view of the sensitivity of the simulated SMB to the bare ice albedo formulation (van Angelen et al., 2012; Tedesco et al., 2016), improving the bare ice albedo representation in MAR should be a priority for the next MAR developments.

Using other reanalyses than ERA-Interim as MAR forcing does not significantly change the comparison with the PROMICE measurements. Table 3 shows the relevance of the ERA-20C temperature correction while the warm biais of 20CRv2 mitigates the cold bias found in $MAR_{ERA-Interim}$. Statistically, $MAR_{ERA-Interim}$ and $MAR_{JRA-55}$ compare the best whereas $MAR_{20CRv2}$ compares the worst.

### 4.2 Surface Mass Balance

As a validation of the SMB simulated by MAR over 1958–2010 (the period covered by seven of the data sets here), we use:

1. The ice core measurements in the accumulation area from Bales et al. (2001, 2009); Ohmura et al. (1999). The MAR accumulations values (here in m W.E./yr) for each of the 246 records are averaged over the years listed in the three





previous references (the mean 1958–2010 is used otherwise) and come from an interpolation of the 4 nearest inverse distance weighted MAR grid cells.

2. The new SMB database available through the PROMICE web portal (http://www.promice.dk) containing a total of ∼ 3000 measurements from 46 sites from 1892 to 2015 and mostly covering the ablation area of the GrIS and local glaciers (Machguth et al., 2016). For each site, the MAR SMB value is corrected in function of the elevation difference given in the PROMICE database and the interpolated MAR 20km topography using a local and time varying SMB vs elevation gradient as explained in Franco et al. (2012). Moreover, the MAR values (here in m W.E.) are an integration of daily MAR outputs over the exact period given for each record in the PROMICE database. The data are not converted in m W.E./yr as some PROMICE records sometimes cover only several months in the melt season. Only the records included in the 1958–2010 period, with an elevation difference with the MAR topography lower than 500m and inside the MAR ice sheet mask are considered here. The comparison is therefore limited to 1616 records here from the PROMICE database (Machguth et al., 2016). Similarly, the same data set has been also used in Noël et al. (2016) for the validation of RACMO2.3.

3. The revised version (fully described in Kjeldsen et al. (2015)) of the 5-km reconstruction of the near-surface air temperature and the land-ice SMB from Box (2013b), hereafter BOX13, spanning 1840–2012 and calibrated to outputs from RACMO2.1/GR forced by ERA-40/ERA-Interim (van Angelen et al., 2011). With respect to the MAR-based reconstructions, this reconstruction is not forced with reanalyses, except the calibration with RACMO2 but is based on in situ observations (Box, 2013b). Absolute uncertainty for the revised SMB estimates from Box (2013b) is estimated by comparison against field data. In situ annual 208 ablation rates over 1985–1992 yield an ablation root-mean-square error of 35% as we can find with RACMO2.1/GR. The comparison with ice-core-derived net accumulation time series from 86 sites shows a 30% accumulation root-mean-square error. A fundamental assumption is that the calibration regression factors, derived over 1960–2012 versus ice cores, meteorological station temperatures, and with RACMO2.1/GR, are stationary in time.

Fig.5c illustrates $MAR_{ERA-Interim}$ SMB validation results. Statistics are listed in Table 4. Correlation exceeds 0.9 and RMSE is ∼40% from 1862 samples within the MAR ice sheet mask over 1958–2010. With respect to MARv2, the accumulation overestimation shown by Vernon et al. (2013) has been partly corrected in MARv3.5.2. Yet, MARv3.5.2 overestimates SMB in the ablation area while MARv2 underestimates it. This results from the bare ice albedo overestimations shown in the previous section for MARv3.5.2. Preliminary results with MARv3.6 where the bare ice albedo varies now between 0.35-0.55 instead 0.4-0.55 allows correcting this overestimation. The bare ice albedo was fixed to 0.45 in MARv2. This shows the impact and importance of improving the bare ice albedo representation in the models, as already stated by van Angelen et al. (2012).

When MAR is forced by reanalyses other than ERA-40/ERA-Interim, we find that i) $MAR_{NCEPv1}$ is most accurate because NCEPv1 is not affected over 1958–1978 by the bias in the humidity assimilation into ERA-40 impacting the MAR precipitation in the non-homogeneous ECMWF time series, ii) the use of ERA-20C-corr partially corrects the SMB overestimation (due to the underestimation of melt) obtained when MAR is forced by unadjusted ERA-20C, iii) $MAR_{20CRv2}$ is more accurate than





$MAR_{20CRv2-corr}$ because the overestimation of melt in $MAR_{20CRv2}$ compensates the SMB overestimation in the ablation area due to albedo overestimation iv) the results of $MAR_{NCEPv2}$ are worse than MAR results using less constrained reanalyses (e.g. 20CRv2) or first generation reanalysis (.e.g. NCEPv1) as a result of the temperature biases in NCEPv2. Moreover, while some of this data have been used in the Box (2013b)'s reconstruction, the comparison of ice core measurements with BOX13

shows the same agreement with MAR (see Table 4). Regarding the comparison with the SMB PROMICE database, the SMB values from BOX13 have been corrected in function of the elevation difference with the PROMICE database as done for MAR. However, for matching the exact period of the PROMICE database, we have simply derived daily values from the monthly BOX13 values by dividing them by the number of days of the month. It is clear that this approximation smooths the melt variability and can be problematic when the period of measurements covers only a few weeks in the melt season explaining

very likely why BOX13 is less correlated with the PROMICE data set than MAR. Finally, it is interesting to note that the comparison with the PROMICE and ice core measurements is quite constant over the whole century (see Table 5) and not better over the recent decades although the number of assimilated data is larger. The lower correlations are reached in the 1950's and 1960's but the number of observations is too limited before 1950 to permit the conclusion that the reliability of the MAR reconstructions are constant in time.

Fig. 6 illustrates how MARv3.5.2 still overestimates snow accumulation for the southern ice sheet in comparison with ice cores (see Fig. 6b) and BOX13 (see Fig. 6c) although this bias has been partly corrected since MARv2 which is wetter in this area than the current MARv3.5.2 (see Fig. 4a). MAR also underestimates accumulation in the south-east versus ice cores but overestimates versus BOX13 because this data set is based on RACMO2 outputs which are known to underestimate accumulation in this area (Noël et al., 2016). In these areas, the spread in the mean 1958–2010 SMB simulated by MAR using

the different reanalyses is below 25 mmWE/yr (see Fig. 6d), confirming that those biases are independent of the forcing used and that improvements in MAR should improve absolute accuracy. Moreover, those biases are in full agreement with the MAR biases found by Koenig et al. (2016) with respect to 2009–2012 airborne snow radar based estimates. In the ablation area (i.e. with respect to the PROMICE database), the MAR biases vary regionally and so systematic biases can not yet be highlighted. Finally, huge differences (> 500 mm/yr) with BOX13 occur along the coastal and mountainous region of the south-east.

MAR underestimates accumulation relatively to BOX13 based on RACMO2. The Polar MM5 model (24km) shows the same underestimation with respect to RACMO2 (Box, 2013b). (Ettema et al. , 2009) attributed the higher accumulation rates in these topographically enhanced precipitation regions to the higher spatial resolution used in RACMO2 (11km). However, MAR simulation at a resolution of 10km (not shown here) does not simulate such an extremely high precipitation and the number of observations in this very wet area is too sparse to confirm the RACMO2 based estimations, suggesting that next

accumulation measurement campaigns should focus on this area.

## 5  Validation with microwave satellite derived melt extent

As in Fettweis et al. (2011b), we use the brightness temperatures collected at K-band horizontal polarization (T19H) to retrieve the daily melt extent from the Scanning Multichannel Microwave Radiometer (SMMR) (1979–1987) and the Special Sensor





Microwave/Imager (SSM/I) (1988–2010) data distributed by the National Snow and Ice Data Center (NSIDC, Boulder, Colorado) (Armstrong et al., 1994, Knowles et al., 2002). A grid cell is considered as melting in MAR (resp. in satellite based data sets) if the daily meltwater production (resp. T19H) is higher than 8 mmWE/yr (resp. 227.5 K). We refer to Fettweis et al. (2011b) for more details about the melt retrieving methodology.

As already presented in Fettweis et al. (2011b), the comparison of the melt extent simulated by MAR and retrieved from the passive microwave satellites is encouraging (see Table 4 and Fig. 5). The RMSE represents ∼30% of the daily variability found in the remote based melt extent over the 1979–2010 summers and correlations are higher than 0.9, regardless of the forcing used. As already shown in the two previous sections, $MAR_{20CRv2}$ (resp. $MAR_{ERA-20C}$) overestimates (resp. underestimates) the melt extent, fully justifying the corrections applied to 20CRv2 and ERA-20C to reduce these biases. Finally, MARv3.5.2

slightly improves the comparison with respect to MARv2 used in Fettweis et al. (2011b).

## 6   Time evolution

.

### 6.1   Temperature

Fig. 7 illustrates the MAR ability to simulate a time series of observed composite near-surface air temperature from Cappelen et al.

(2014). As the latter is based on coastal weather station measurements of South and West Greenland, a large part of the interannual variability comes from SST changes which are prescribed every 6 hours into MAR. The remaining part comes from changes in the general circulation (Fettweis et al., 2013a), also prescribed at the MAR lateral boundaries. Therefore, this section rather evaluates the ability of the different MAR forcings to represent the observed temperature variability. As these observations have been assimilated into BOX13, this reconstruction perfectly matches the observations.

With respect to the 1980–2010 average, the 1900–1920 coastal temperatures were lower and with a low interannual variability which is only well represented by $MAR_{20CRv2-corr}$ and $MAR_{20CRv2c}$ although these simulations underestimate the negative temperature anomalies observed during this period after BOX13. A first maximum of temperature was reached in 1930 which is only well represented by $MAR_{20CRv2c}$. $MAR_{ERA-20C-corr}$ simulates this maximum earlier while $MAR_{20CRv2c}$ underestimates it. This maximum is also observed in the summer (JJA) time series but underestimated in all of the MAR based

time series. After this optimum warm period discussed in Chylek et al. (2006), there are two minor temperature maxima in the beginning of the 1960's and at the end of the 1970's overestimated by $MAR_{20CRv2c}$ and $MAR_{NCEPv1}$ and underestimated by the MAR time series using the other forcings. Temperature differences of several degrees between the JRA forced time series before and during the satellite era (starting at the end of the 1970's) suggest biases in the JRA-based SST before 1980. Except in $MAR_{20CRv2-corr}$ (and in $MAR_{ERA-Interim}$ to a lesser extent) where the temperature variability is very smoothed,

the summer and annual warming in the 1990's and 2000's is well represented in all the MAR-based times series.

    When integrated over the whole ice sheet (see Fig. 7c), all MAR reconstructions show a decrease of the summer mean temperature (gauging the melt) after 1930 until the beginning of the 1990's when an abrupt temperature increase of ∼2°C in



10 years is simulated. Before 1930, the MAR reconstructions diverge although the reanalyses are supposed to represent the same climate variability. However, the comparison with BOX13 constrained by DMI coastal weather station measurements is the closest when MAR is forced by 20CRv2(c) because SST is assimilated into 20CRv2(c) but not into ERA-20C. Finally, as absolute temperatures are shown here, we can see that MAR is systematically 0.5-1°C colder than BOX13 as a result of the

MAR cold bias discussed in Section 4.1.

## 6.2 Surface Mass Balance

Time series of the SMB components integrated over the whole GrIS are presented in Fig. 8. Before 1930, as for the JJA mean GrIS near-surface temperature (see Fig. 7), there are large discrepancies between the MAR based run-off reconstructions suggesting that large improvements (i.e. assimilating more data) are still needed in the reanalysis before this period. After

the warm period observed in the 1930's (Chylek et al., 2006), all the MAR reconstructions suggest an increasing SMB due to heavier snowfall and less melt. Regarding the period 1960–1990, the meltwater run-off amount is low and stable. The highest SMB occurred in the 1970's while they are some discrepancies among the models. This maximum is the highest when MAR is forced by ERA-40 which is also used to force RACMO on which BOX13 is based. At the beginning of this century, all the models simulate a SMB decrease as a result of an increasing surface melt, reaching a record minimum on or after 2010.

A second SMB minimum is simulated around 1930 by MAR as a results of high melt and low accumulation. This minimum is less pronounced in BOX13 which includes considerable smoothing by the weighted averaging of annual core and monthly station temperature values. Therefore, BOX13 may suffer from more damping than what MAR can produce with 6-hourly forcings. Finally, MAR suggests a significant snowfall increase from 1900–1920 to 1950 in opposition to Hanna et al. (2011). However, such an increase is also suggested in the Box's reconstruction (Box et al., 2013a) and in the ice cores (Mernild et al.,

2015) but in a less extent than in the MAR simulations (see Fig.9). Part of the MAR simulated snowfall increase may be caused by an artificial increase of the daily sea level pressure variability over 1900–1950 (see Fig. 9d) and the associated strengthened eddy activity. The 20CRv2c reanalysis is an ensemble mean suggesting that lower the amount of assimilated data is, higher the spread is for a given event. This smooths the pressure fields and therefore decreases the amount of humidity advected into the MAR free atmosphere and then the precipitation rate simulated by MAR although the 20CRv2 reanalysis itself simulates higher

precipitation during this period (Hanna et al., 2011). ERA-20C is not an ensemble mean but it is likely that a lower amount of assimilated data could also induce smoother pressure fields although ERA-20C seems to suggest that the storm activity were higher at the beginning of the last century than in the 1920–1940 period. Therefore, this apparent significant precipitation increase from 500 Gt/yr to > 600 Gt/yr simulated by MAR over 1900–1950 should be considered with caution since both reanalyses forced MAR simulation disagree in the location where this increase takes place (western coast vs eastern coast) and

that a part of this increase could just be due to artefact in 20CRv2(c). Finally, it is interesting to note that (Hanna et al., 2016) showed also an increase of the variability of the 20CRv2c-based Greenland Blocking Index (GBI) through the last century.

We can see in Fig. 9 that the pattern of snowfall increase over 1921–1950 is quite different following the reconstruction and that there are some disagreements with the ice core based trend listed in Mernild et al. (2015). $MAR_{20CRV2c}$ suggests a decrease of accumulation along the west coast and a significant increase along the eastern coast with the highest increase at





the south-east as the other reconstructions. $MAR_{ERA-20C-corr}$ suggests a decrease only at the north of the ice sheet and a significant increase along the western coast in disagreement with the two other reconstructions. Finally, BOX13 suggests an increase only at the south(-east) of the ice sheet. The decrease seen in the ice cores in the Humboldt-NEEM area (at the north-west) is well represented by the 3 reconstructions but they fail to simulate the decrease observed at D1 near Tasiiliaq. The other

ice cores suggests rather a positive trend in agreement with all the reconstruction but MAR mostly overestimates the observed trend while BOX13 is in better agreement with ice cores. The significant accumulation increase simulated by $MAR_{20CRV2c}$ along the north-eastern coast and simulated by $MAR_{ERA-20C-corr}$ along the western coast seems to be overestimated with respect to ice core measurements. Unfortunately, no gauge observation is available along the south-eastern coast to confirm the significant snowfall increase simulated by the 3 reconstructions in this area over 1921–1950.

## 7  Discussion and conclusions

Reconstructions of the GrIS SMB from the beginning of the last century (1900–2015) were carried out using the regional climate model MAR forced by 8 reanalyses. Over the recent decades, all MAR time series compares very well with in situ measurements, ice core and satellite derived melt extent while temperature corrections were needed in the 20CRv2(c) and ERA-20C reanalysis at the MAR boundaries. MAR forced by ERA-Interim shows the best comparison with observations for

1980 onward while NCEP-NCARv1 outperforms ERA-40/ERA-Interim since the 1950's.

Around 1930, all the reconstructions agree on a SMB minimum concurrent with the warm period observed in the coastal temperatures (Chylek et al., 2006). Afterwards, the reconstructions suggest after a melt decrease until the 1970's and an accumulation increase until the middle of the 1940's. A second minimum of SMB occurs in the 1960's when a minimum of accumulation is reached while the highest SMB rates are reached over the 1970–1990's as a consequence of lower melt and

higher accumulation than before. From the end of the 1990's, all the reconstructions show a significant SMB decrease resulting from a surface melt increase to the 2010's when the absolute minimum of SMB is reached in all time series from 1900.

There are however large discrepancies before the 1930's between the MAR reconstructions as well as with the Box (2013b) time series. MAR forced by ERA-20C suggests a continuous run-off increase from the 1900's to 1930's while MAR forced by 20CRv2(c) and, in a less extent, BOX13 suggest a run-off decrease from the 1900's to the 1920's followed by a melt increase

reaching a first maximum at the beginning of the 1930's. The same time discrepancies can be seen in the MAR simulated near-surface temperatures. MAR simulates also a significant snowfall increase from the 1910's to the 1940's. The reconstruction from Box et al. (2013a) and ice cores (Mernild et al., 2015) suggest also an accumulation increase over this period but in a less extent than MAR while Hanna et al. (2011) suggested the opposite. Long term ice core data facilitate validation of an overall ice sheet snowfall increase in the first half of the last century and the comparison with MAR is good where a few ice cores are

available. This increase is however bracketed in several ice cores in the dry north as well simulated by MAR but not for the single core (see D1 in Fig. 9) in the south-east showing decreasing snowfall. Thus, the ice sheet averaged core values are then almost insignificant while MAR suggests a significant increase along the south-east coastal ridge where ice cores are missing. This suggests that new ice core drilling's are needed in this area to confirm the MAR accumulation increase. Moreover, this





accumulation increase in MAR coincides with an increase of the daily sea level pressure variability in forcing reanalyses impacting the amount of humidity advected into the MAR integration domain. The 20CRv2(c) reanalysis is an ensemble mean of 56 members suggesting that lower the amount of assimilated data is, smoother the pressure fields are. Therefore, the increase of the daily sea level pressure variability could just be an artefact coming from forcing reanalysis. While ERA-20C is not an

ensemble mean, MAR forced by this reanalysis shows the same precipitation increase but not on the same locations than MAR forced by 20CRv2(c). On the other hand, the amount of assimilated data into ERA-20C is lower during this period. Therefore, without enough gauge observations, it is hard to conclude that this MAR-based significant accumulation increase along the south-east coastal ridge over the first half of last century is robust or whether it is just an artefact coming from the forcing reanalyses (which need to be more constrained to be in agreement before the 1930's). Belleflamme et al. (2013) already

showed large discrepancies in the general circulation over Greenland simulated by those two reanalyses before 1940 explaining the significant differences in the simulated run-off and snowfall variability.

The period 1961–1990 has been considered as period when the total mass balance of the Greenland ice sheet was stable (Rignot and Kanagaratnam, 2006) and near zero. However, at the last century scale, all MAR reconstructions suggest that SMB was particularly positive during this period (SMB was the most positive from the 1970's to the middle of the 1990's)

suggesting that mass gain could rather occur during this period in agreement with results from Colgan et al. (2015).

Finally, with respect to the 1961–1990 period, the integrated contribution of the GrIS SMB anomalies over 1900–2010 is a sea level rise of about 15±5 mm with a null contribution from the 1940's to the 2000's suggesting that the recent contribution of GrIS to sea level change (van den Broeke et al., 2016) are unprecedented in the last century GrIS history. A next step to evaluate total mass changes should be to force ice sheet models with these MAR reconstructions to confirm the stability of the

ice dynamics over 1961–1990 and to better understand the recent acceleration of ice dynamics (van den Broeke et al., 2016). This recent acceleration of ice dynamics could partly result from the purge of the extra mass (accumulated through the 3 last decades of last century with respect to previous decades) enhanced by the recent melt increase lubricating the glaciers - bedrock interface.

## 7.1 Data availability

All MARv3.5.2 outputs presented here are available at ftp://ftp.climato.be/fettweis/MARv3.5/Greenland/ and the code source of MARv3.5.2 is available at ftp://ftp.climato.be/fettweis/MARv3.5/.src/. The ECMWF reanalyses (ERA-Interim, ERA-40 and ERA-20C) has been downloaded from http://apps.ecmwf.int/datasets/. The NCEP-NCARv1, NCEP-NCARv2 and the 20CRv2(c) reanalyses come from http://www.esrl.noaa.gov/psd/data/ while the JRA55 reanalysis comes from https://climatedataguide.ucar The brightness temperature used to retrieve the melt extent from the satellite were downloaded from http://nsidc.org/. Finally,

the PROMICE data used to validate MAR are available at http://www.promice.dk/.

*Acknowledgements.* The PROMICE (Programme for Monitoring of the Greenland Ice Sheet) network is funded by the Danish Energy Agency (DANCEA) programme; data are available at http://www.promice.dk. Computational resources have been provided by the Consortium des Équipements de Calcul Intensif (CECI), funded by the Fonds de la Recherche Scientifique de Belgique (F.R.S.-FNRS) under Grant





No. 2.5020.11 and the Tier-1 supercomputer (Zenobe) of the Fédération Wallonie-Bruxelles, infrastructure funded by the Walloon Region under the grant agreement n°1117545. Finally, Xavier Fettweis is a Research Associate from the Fonds de la Recherche Scientifique de Belgique (F.R.S.-FNRS). J. Box was supported by the Danish Research Council research grant FNU 4002-00234.





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





**Figure 1.** a) In background, the interannual variability (i.e. standard deviation) of the JJA mean 700 hPa temperature (T700) simulated by ERA-Interim over 1980-1999. Units are Celsius degrees. The contours of the mean JJA T700 is plotted in dashed blue. b)-h) Mean anomalies of the JJA 700 hPa temperature simulated by the different reanalyses used in this study with respect to ERA-Interim over 1980–1999 (in Celsius degrees). No comparison is shown above 2000m a.s.l. in the aim of showing comparisons in the free atmosphere (700hPa) only and the data sets are shown here by using their native Lat-Long projection.





**Figure 2.** Idem as Fig. 2 but for the mean annual geopotential height (Z500) at 500hPa over 1980–1999. Units are meters.





**Figure 3.** a) Difference between the mean annual SMB (in mmWE/yr) simulated by MARv2 forced by ERA-Interim and MARv3.5.2 forced by ERA-Interim over 1980–1999. b) Difference between the mean 1980–1999 annual SMB simulated by MARv3.5.2 forced by ERA-40 and simulated by MARv3.5.2 forced by ERA-Interim. c) Idem as b) but for ERA-20C. c) Idem as b) but for ERA-20C-corr. e) Idem as b) but for JRA-55. f) Idem as b) but for NCEPv1. g) Idem as b) but for NCEPv2. h) Idem as b) but for 20CRv2. i) Idem as b) but for 20CRv2-corr. j) Idem as b) but for 20CRv2c. Finally, the areas where the differences are lower than the interannual variability of MARv3.5.2 forced by ERA-Interim over 1980–1999 are hatched.





**Figure 4.** Same as Fig. 3 but for snowfall (in mmWE/yr).





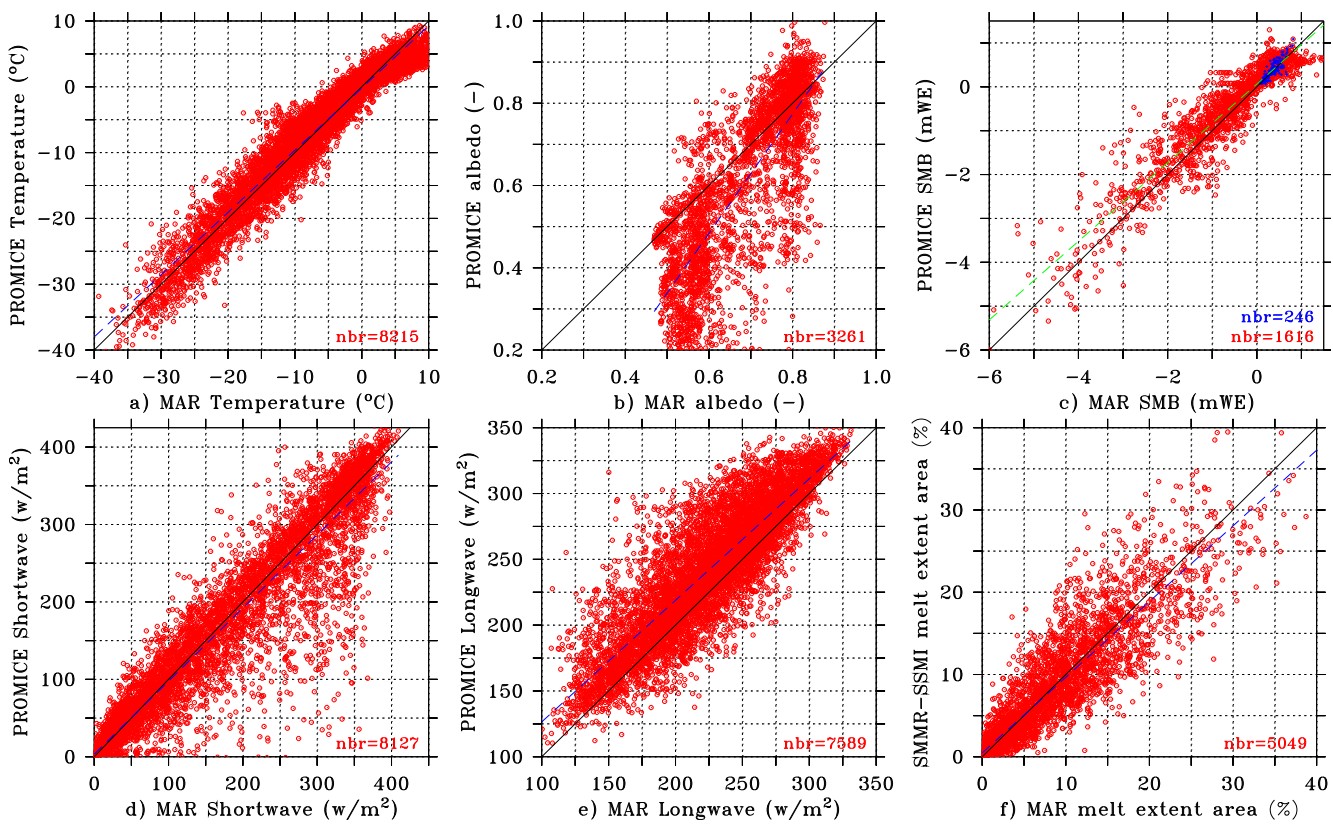

**Figure 5.** a) Scatter plot of the MAR$_{\mathrm{ERA-Interim}}$ daily near-surface temperature versus near-surface daily temperature recorded by 12 AWS's from the PROMICE network over 2008-2010. The number of observations used here is listed in red and units are Celsius degrees. b) Same as a) but for the surface albedo. c) Scatter plot of the MAR$_{\mathrm{ERA-Interim}}$ SMB (in mWE) with respect to ice core measurements in the accumulation area (in blue) and SMB measurements in the ablation (in red) from the PROMICE data set over 1958–2010. We refer to the text for more details on how this comparison is performed. d) Same as a) for the shortwave downward radiative flux (in W/m$^2$). e) Same as d) for the longwave downward radiative flux. f) Daily melt extent (in % of the ice sheet area) simulated by MAR$_{\mathrm{ERA-Interim}}$ over the 1979–2010 summers (May-September) versus the satellite derived one. More information about the thresholds used for retrieving the melt extent is given in the text.





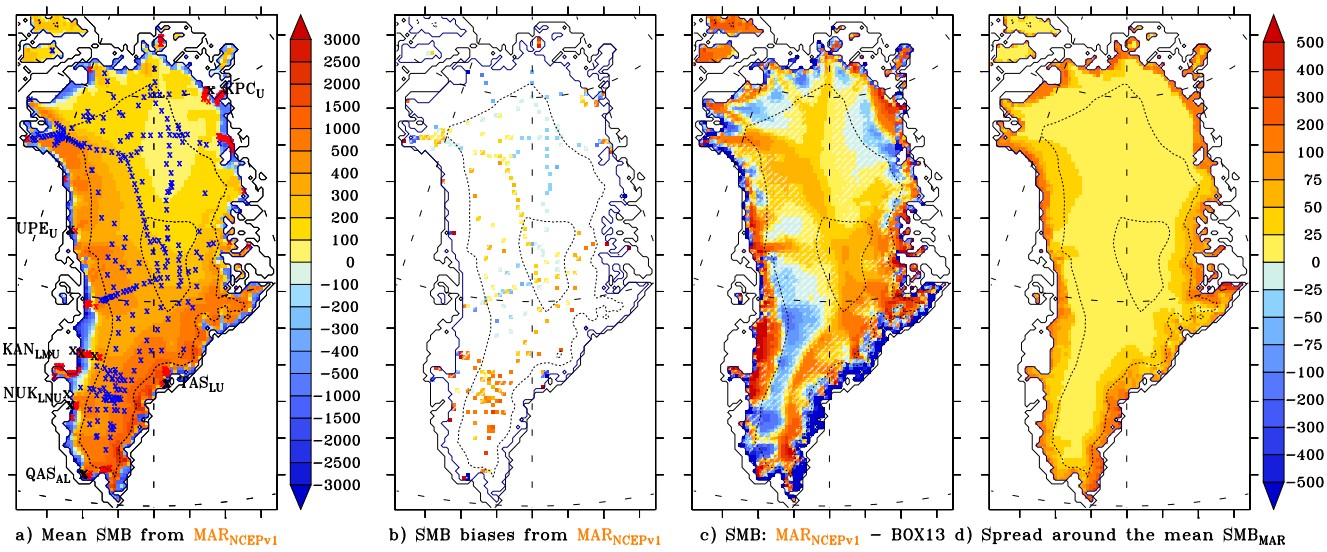

**Figure 6.** a) Mean annual SMB (in mmWE/yr) simulated by MAR forced by NCEPv1 over 1958–2010. The ice core locations from Bales et al. (2001, 2009); Ohmura et al. (1999) used to validate MAR are quoted in blue while the PROMICE SMB sites (Machguth et al., 2016) are in red. b) Mean biases (in mmWE/yr) of MAR forced by NCEPv1 over 1958–2010 with respect to both ice core and PROMICE based SMB estimations. The biases lower than the interannual variability of MARv3.5.2 forced by NCEPv1 over 1958–2010 are hatched. c) Comparison over 1958–2010 between the mean SMB (in mmWE/yr) simulated by MAR and from the Box's reconstruction. Again, the biases lower than the interannual variability of MARv3.5.2 forced by NCEPv1 over 1958–2010 are hatched. d) Spread (i.e. standard deviation in mmWE/yr) around 6 estimations of the mean 1958–2010 SMB as simulated by MAR forced by ERA, NCEPv1, JRA, ERA-20C-corr, 20CRv2-corr, 20CRv2c.





**Figure 7.** a) Time series of the annual SW Greenland near-surface temperature (built by merging series of the Ilulissat, Nuuk and Qaqortoq coastal weather station from the Danish Meteorological Institute (DMI)) as observed (in brown) according to Cappelen et al. (2014), retrieved from the BOX13 reconstruction (in black) and as simulated by MAR with the different forcings. Values are anomalies with respect to 1980–2010 and 10yr-runnings are shown. b) Same as a) for the summer (JJA) SW Greenland near-surface temperature. c) Mean GrIs summer (JJA) near-surface temperature (in °C) as simulated by MAR using the different forcings. The ERA-20c (without temperature correction) forced MAR time series as well as the BOX13 reconstruction based time series are also shown.





**Figure 8.** Top) Time series of the annual SMB (in GT/yr) integrated over the whole ice sheet as simulated by MAR using the different listed forcings and coming from the Box's reconstruction (BOX13). Middle) Same as Top) but for snowfall. Bottom) Same as Top) but for run-off. Finally, 10yr-running means only are shown for both Middle and Bottom for more readability.





**Figure 9.** a) Annual snowfall trend (in mmWE/yr$^2$) over 1921–1950 as simulated by MAR forced by 20CRv2c. The observed trend (in mmWE/yr$^2$) from some locations listed in Mernild et al. (2015) are also given. The negative (resp. positive) trends are printed in blue (resp. in red). Trend of total precipitation (rain+snow) are also listed in black for 5 coastal weather stations from DMI. b) Same as a) but for MAR$_{ERA-20C-corr}$. c) Same as a) but for BOX13. d) Time series of the annual mean daily variability (i.e. standard deviation of the daily values) of the sea level pressure around Greenland ($0° W \leq$ longitude $\leq 80° W$ and $55° N \leq$ latitude $\leq 85° N$) from 20CRv2c (in green), ERA-20C (in red) and NCEPv1 (in orange). The ensemble mean spread (i.e. the standard deviation of the ensemble deviations at each time) from 20CRv2c over the same area is also plotted in dash. Finally, only 10yr-running means are shown for more readability.





**Table 1.** Average and standard deviation (gauging the inter-annual variability) of the annual SMB components simulated by MAR over 1980–1999 and from the Box's reconstruction (interpolated to the MAR 20km grid). Units are $GT\ yr^{-1}$ and acronym of each simulation ($RCM_{forcings}$) is given in the first column. The surface mass balance (SMB) equation is here SMB = snowfall + rainfall − run-off − water fluxes. The run-off is the part of not refrozen water from both surface melt and rainfall reaching the ocean. Finally, the free atmosphere temperature from the forcing with an asterisk were corrected according to Section 2.2

| Simulation acronym | SMB | Snowfall | Rainfall | Run-off | Water fluxes | Meltwater |
|---|---|---|---|---|---|---|
| $MAR_{ERA-Interim}$ | $480 \pm 87$ | $683 \pm 56$ | $28 \pm 5$ | $220 \pm 52$ | $12 \pm 4$ | $427 \pm 82$ |
| $MAR_{ERA-40}$ | $529 \pm 89$ | $716 \pm 57$ | $31 \pm 6$ | $210 \pm 54$ | $9 \pm 3$ | $418 \pm 86$ |
| $MAR_{ERA-20C}$ | $500 \pm 71$ | $624 \pm 76$ | $18 \pm 4$ | $126 \pm 35$ | $15 \pm 3$ | $296 \pm 59$ |
| $MAR_{ERA-20C-corr}$ | $491 \pm 84$ | $665 \pm 59$ | $26 \pm 6$ | $190 \pm 48$ | $10 \pm 3$ | $399 \pm 77$ |
| $MAR_{NCEPv1}$ | $467 \pm 88$ | $675 \pm 59$ | $28 \pm 6$ | $228 \pm 53$ | $8 \pm 4$ | $440 \pm 82$ |
| $MAR_{NCEPv2}$ | $486 \pm 86$ | $672 \pm 60$ | $22 \pm 5$ | $200 \pm 46$ | $8 \pm 4$ | $409 \pm 74$ |
| $MAR_{20CRv2}$ | $420 \pm 102$ | $703 \pm 60$ | $31 \pm 6$ | $221 \pm 52$ | $12 \pm 2$ | $432 \pm 82$ |
| $MAR_{20CRv2-corr}$ | $459 \pm 88$ | $670 \pm 57$ | $22 \pm 4$ | $309 \pm 67$ | $5 \pm 5$ | $559 \pm 102$ |
| $MAR_{20CRv2c}$ | $456 \pm 92$ | $680 \pm 59$ | $25 \pm 6$ | $241 \pm 63$ | $8 \pm 4$ | $462 \pm 97$ |
| $MAR_{JRA-55}$ | $482 \pm 88$ | $670 \pm 57$ | $29 \pm 5$ | $209 \pm 52$ | $9 \pm 4$ | $412 \pm 83$ |
| Box (2013) | $502 \pm 74$ | $735 \pm 62$ | | $229 \pm 47$ | | $424 \pm 71$ |





**Table 2.** Localisation (Latitude, longitude and elevation) of the 12 AWS's from the PROMICE network used here to validate MAR. The location of the corresponding MAR grid cell is also listed.

| AWS | PROMICE | | | MAR | | |
|---|---|---|---|---|---|---|
| | Lat. (° N) | Lon. (° E) | Elev. (m) | Lat. (° N) | Lon. (° E) | Elev. (m) |
| KAN_L | 67.10 | -49.93 | 680 | 67.08 | -49.92 | 649 |
| KAN_M | 67.07 | -48.82 | 1270 | 67.08 | -48.78 | 1298 |
| KAN_U | 67.00 | -47.02 | 1850 | 66.98 | -46.93 | 1896 |
| KPC_U | 79.83 | -25.12 | 870 | 79.85 | -25.07 | 766 |
| NUK_L | 64.48 | -49.53 | 560 | 64.49 | -49.54 | 877 |
| NUK_N | 64.95 | -49.88 | 930 | 64.94 | -49.90 | 888 |
| NUK_U | 64.50 | -49.26 | 1140 | 64.50 | -49.26 | 1117 |
| QAS_A | 61.24 | -46.73 | 1009 | 61.30 | -46.75 | 1075 |
| QAS_L | 61.03 | -46.85 | 310 | 61.04 | -46.83 | 525 |
| TAS_L | 65.64 | -38.90 | 270 | 65.64 | -38.91 | 440 |
| TAS_U | 65.70 | -38.87 | 580 | 65.73 | -38.90 | 655 |
| UPE_U | 72.89 | -53.53 | 980 | 72.89 | -53.54 | 1001 |



**Table 3.** Mean correlation, bias, RMSE and correlation over the 12 AWS's listed in Table 2 between MAR forced by the different reanalyses and daily observations from the PROMICE network over 2008–2010. Statistics are given for the surface pressure (SP), near-surface temperature (TAS) over the whole year and the summer month only (JJA), shortwave downward flux (SWD) and longwave downward flux (SWD)

| Simulation acronym | SP | TAS (°C) | | | Summer TAS (°C) | | |
|---|---|---|---|---|---|---|---|
| | CORR | BIAS | RMSE | CORR | BIAS | RMSE | CORR |
| MAR$_{ERA-Interim}$ | 0.99 | -0.29 | 2.32 | 0.96 | -0.65 | 2.38 | 0.95 |
| MAR$_{ERA-20C}$ | 0.99 | -1.04 | 2.78 | 0.95 | -1.42 | 2.92 | 0.93 |
| MAR$_{ERA-20C-corr}$ | 0.99 | -0.26 | 2.56 | 0.95 | -0.61 | 2.64 | 0.93 |
| MAR$_{NCEPv1}$ | 0.99 | -0.04 | 2.48 | 0.95 | -0.26 | 2.47 | 0.93 |
| MAR$_{NCEPv2}$ | 0.99 | -0.19 | 2.52 | 0.95 | -0.44 | 2.51 | 0.93 |
| MAR$_{20CRv2}$ | 0.98 | 0.30 | 3.16 | 0.92 | -0.27 | 3.07 | 0.90 |
| MAR$_{20CRv2-corr}$ | 0.98 | -0.42 | 3.21 | 0.92 | -1.02 | 3.25 | 0.89 |
| MAR$_{20CRv2c}$ | 0.98 | -0.33 | 3.09 | 0.93 | -0.76 | 3.05 | 0.91 |
| MAR$_{JRA-55}$ | 0.99 | -0.56 | 2.51 | 0.96 | -1.08 | 2.62 | 0.94 |
| MARv2$_{ERA-Interim}$ | 0.99 | -0.98 | 2.73 | 0.95 | -1.39 | 2.90 | 0.94 |

| Simulation acronym | SWD (W/m$^2$) | | | LWD (W/m$^2$) | | |
|---|---|---|---|---|---|---|
| | BIAS | RMSE | CORR | BIAS | RMSE | CORR |
| MAR$_{ERA-Interim}$ | 3.42 | 27.07 | 0.96 | -16.92 | 28.13 | 0.84 |
| MAR$_{ERA-20C}$ | 4.05 | 30.54 | 0.96 | -19.98 | 32.35 | 0.79 |
| MAR$_{ERA-20C-corr}$ | 3.17 | 30.43 | 0.96 | -16.33 | 30.29 | 0.79 |
| MAR$_{NCEPv1}$ | 1.84 | 29.58 | 0.96 | -14.19 | 29.64 | 0.79 |
| MAR$_{NCEPv2}$ | 2.70 | 29.74 | 0.96 | -14.64 | 30.10 | 0.79 |
| MAR$_{20CRv2}$ | 1.75 | 33.51 | 0.95 | -14.28 | 32.55 | 0.74 |
| MAR$_{20CRv2-corr}$ | 0.21 | 32.30 | 0.95 | -14.34 | 32.54 | 0.74 |
| MAR$_{20CRv2c}$ | 0.73 | 32.21 | 0.95 | -14.28 | 32.55 | 0.74 |
| MAR$_{JRA-55}$ | 3.71 | 26.92 | 0.96 | -17.98 | 29.41 | 0.83 |
| MARv2$_{ERA-Interim}$ | -1.8 | 27.64 | 0.95 | -19.52 | 31.42 | 0.81 |



**Table 4.** Comparison with SMB from the PROMICE database over 1958–2010, ice core based accumulation from Bales et al. (2001, 2009); Ohmura et al. (1999) and satellite derived melt extent over 1979–2010. $MAR_{ERA-Interim}$ (resp. $MAR_{ERA-40}$) means that MAR was forced by ERA-40 over 1958–1978 (resp. 1958–2000) and ERA-Interim over 1979–2015 (resp. 2001–2015). Finally, $MAR_{ERA-Interim}*$ means the extrapolation of Franco et al. (2012) was not used to correct the SMB of the elevation difference between MAR and the PROMICE measurement sites.

| Simulation acronym | SMB - PROMICE (m W.E.) | | | Accumulation (m W.E./yr) | | | Melt extent (%) | | |
|---|---|---|---|---|---|---|---|---|---|
| | BIAS | RMSE | CORR | BIAS | RMSE | CORR | BIAS | RMSE | CORR |
| $MAR_{ERA-Interim}$ | +0.14 | 0.46 | 0.93 | +0.02 | 0.08 | 0.91 | +0.0 | 2.8 | 0.93 |
| $MAR_{ERA-40}$ | +0.20 | 0.48 | 0.93 | +0.03 | 0.09 | 0.91 | -0.1 | 2.9 | 0.92 |
| $MAR_{ERA-20C}$ | +0.39 | 0.67 | 0.91 | -0.03 | 0.07 | 0.91 | -2.0 | 3.8 | 0.90 |
| $MAR_{ERA-20C-corr}$ | +0.22 | 0.52 | 0.93 | +0.01 | 0.07 | 0.91 | -0.4 | 3.0 | 0.91 |
| $MAR_{NCEPv1}$ | +0.13 | 0.45 | 0.93 | +0.03 | 0.09 | 0.92 | +0.2 | 2.9 | 0.92 |
| $MAR_{NCEPv2}$ | +0.26 | 0.52 | 0.93 | +0.03 | 0.09 | 0.92 | -0.3 | 2.9 | 0.92 |
| $MAR_{20CRv2}$ | +0.01 | 0.47 | 0.93 | +0.01 | 0.08 | 0.92 | +2.0 | 4.5 | 0.92 |
| $MAR_{20CRv2-corr}$ | +0.18 | 0.50 | 0.92 | +0.01 | 0.08 | 0.92 | +0.1 | 3.4 | 0.91 |
| $MAR_{20CRv2c}$ | +0.14 | 0.49 | 0.92 | +0.02 | 0.09 | 0.90 | +0.6 | 3.7 | 0.91 |
| $MAR_{JRA-55}$ | +0.18 | 0.48 | 0.93 | +0.01 | 0.07 | 0.92 | -0.2 | 2.8 | 0.92 |
| BOX13 | +0.16 | 0.68 | 0.84 | +0.00 | 0.08 | 0.92 | | | |
| $MARv2_{ERA-Interim}$ | -0.08 | 0.58 | 0.90 | +0.06 | 0.14 | 0.82 | +0.1 | 2.9 | 0.91 |
| $MAR_{ERA-Interim}*$ | +0.34 | 0.74 | 0.86 | | | | | | |



**Table 5.** Same as Table 4 but for each decade over 1910–2010. The numbers of observations (nbr) as well as the standard deviation (std) of observations are also listed.

| Decade | nbr | std | $MAR_{ERA-20C-corr}$ | | | $MAR_{20CRv2c}$ | | | BOX13 | | |
|---|---|---|---|---|---|---|---|---|---|---|---|
| | | | BIAS | RMSE | CORR | BIAS | RMSE | CORR | BIAS | RMSE | CORR |
| 1910's | 12 | 0.73 | 0.03 | 0.20 | 0.97 | -0.02 | 0.18 | 0.97 | 0.16 | 0.35 | 0.95 |
| 1920's | 19 | 1.13 | 0.03 | 0.41 | 0.94 | -0.02 | 0.29 | 0.97 | 0.66 | 1.04 | 0.84 |
| 1930's | 27 | 1.13 | -0.04 | 0.23 | 0.98 | -0.22 | 0.43 | 0.97 | 0.43 | 0.83 | 0.98 |
| 1940's | 45 | 0.11 | -0.01 | 0.04 | 0.92 | -0.03 | 0.05 | 0.90 | -0.03 | 0.05 | 0.93 |
| 1950's | 274 | 0.68 | 0.10 | 0.45 | 0.77 | -0.13 | 0.55 | 0.76 | 0.11 | 0.49 | 0.78 |
| 1960's | 107 | 0.49 | 0.16 | 0.39 | 0.71 | 0.12 | 0.40 | 0.78 | 0.02 | 0.32 | 0.78 |
| 1970's | 162 | 1.07 | -0.11 | 0.35 | 0.96 | -0.17 | 0.40 | 0.97 | 0.39 | 0.64 | 0.88 |
| 1980's | 1072 | 1.03 | 0.02 | 0.42 | 0.91 | -0.06 | 0.46 | 0.91 | 0.16 | 0.56 | 0.85 |
| 1990's | 452 | 1.09 | 0.25 | 0.47 | 0.93 | 0.11 | 0.43 | 0.93 | 0.01 | 0.47 | 0.90 |
| 2000's | 210 | 1.52 | 0.08 | 0.64 | 0.91 | -0.12 | 0.62 | 0.92 | 0.28 | 1.16 | 0.68 |