# Peer review of "Reconstructions of the 1900–2015 Greenland ice sheet surface mass balance using the regional climate MAR model"

_The Cryosphere, 2016_

## Referee Comment (RC1) · Anonymous Referee #1 · 27 Dec 2016

Review of

Reconstructions of the 1900–2015 Greenland ice sheet surface mass balance using the regional climate MAR model

By Fettweis and others

General assessment

This paper describes output of the regional climate model MAR, at 20 km horizontal resolution, over the Greenland ice sheet, forced at the lateral boundaries and the model top by a multitude of global reanalysis products, some if which go as far back in time as the 19th century, but in this paper the period 1900 – present is addressed. The paper also evaluates model performance for the benchmark run forced with ERA-interim. These results are certainly interesting and potentially important, but I have two main concerns. First, I found the paper very hard to read, mainly because of the writing style and the fact that the authors decided to show nearly all results rather than making an informed decision on the most relevant parameters and results. Second, the results are described in a merely qualitative fashion, without a more quantitative analysis of the reasons of the substantial differences between the various model runs. As a result a clear conclusion is lacking as to the quality of the re-analysis products for reproducing Greenland surface mass balance. These concerns are described in more detail below, together with some textual comments where clarification is needed (listing not complete). Major revisions will be necessary to bring the paper to a publishable level.

Major comments

The paper is very difficult to read, owing to the multitude of model acronyms and the overwhelming number of figure frames and tables contents. Already early in the paper I got confused by all MAR model versions presented. On page 3 alone, there is mention of MAR, MAR3.5.2, MARv2, MARv3.x, MARv3.2. . ... Some more sobering statistics: the manuscript contains six tables with about 500 numbers, sixty lines in line graphs, more than 40 maps, and the acronym MAR is used almost 300 times!

Readability can be further improved by not combining multiple results in a single sentence. Moreover, results should be easily traceable in figures. This now is not always the case. For example, p. 9, l. 17 reads: "MAR also underestimates accumulation in the south-east versus ice cores but overestimates versus BOX13 because this data set is based on RACMO2 outputs which are known to underestimate accumulation in this area (Noël et al., 2016)." This sentence starts by stating that MAR underestimates accumulation in the southeast compared to ice cores. But in Fig. 6c I see red colours, indicating an overestimation? The sentence continues that MAR overestimates (accumulation) vs. BOX 13, but Fig. 6c shows red and blue colours? Then it concludes that BOX13 is wrong in this area because RACMO2 outputs are underestimating accumu-

Interactive
comment

lation in the southeast. So what is the conclusion?

p. 3, l. 13: how can SMB be robustly calculated outside the ice sheet mask? If ice is assumed to be present in a certain grid cell that is currently tundra, then the local climate would be altered (cooled) and hence SMB would be influenced?

p. 3: Selecting the 700 hPa temperature as predictor for melt appears unfortunate, because this pressure level intersects with the ice sheet surface. This implies that T at 700 hPa at the lower parts of the ice sheet represents a free atmosphere temperature, at intermediate parts of the ice sheet it represents a boundary layer value (temperature inversion) and at the highest levels a below-surface (extrapolated?) value. This is confirmed by the blue lines in Fig. 1a, which show a conspicuous local minimum over the higher parts of the ice sheet, clearly caused by the ice sheet surface, a feature that would not be expected if a higher level (500 hPa) had been selected. The authors partly recognize this problem by masking out the level below 2000 m in Fig. 1, but that is again unfortunate because melt also does take place above this altitude so part of the interesting information is lost.

p. 4: Why using the obsolete 20CRv2 at all when an improved/corrected product is available (20CRv2c)?

p. 5: A temperature correction of +/- 1 C is applied to 20CRv2 and ERA-20C, based on a 1980-1999 comparison with ERA-Interim. Can one be sure that these biases have been constant before and after this period? Can you show how stable this bias has been in time for the full period 1958-2015 for which ERA-Interim and ERA-40 are available, which appear to perform adequately? Why is this only done for NCEPv1 (line 26)? Why are these results, which are instrumental for the interpretation of the results in this paper, not shown? The same question applies to the height of the 500 hPa level displayed in Fig. 2. How constant are these biases in time over the period with reliable forcing ($\sim$1950-present)?

p. 7: To improve the logical sequence of the paper, the evaluation of the ERA-Interim
forced run (section 4) should be presented before the other results in Figs. 1-3.

Figure 5: Knowing that absorbed shortwave radiation is the main energy source for melting, it is quite remarkable that ablation is so well reproduced in Fig. 5c, while surface albedo is clearly too high in MAR, by up to a factor of 2. This would imply that, all other things being equal, ablation would be significantly overestimated had albedo been correct. How can this be reconciled, are there compensating errors?

Textual comments

p. 1, l. 9: validated -> support

p. 1, l. 13: The period 1961-1990 is not commonly chosen as reference period because there was approximate balance, rather it is the only official climatological period that can be chosen before significant changes occurred in Greenland.

p. 1, l. 19: "stationarity assumption" Unclear, please explain.

p. 1, l. 20: "…only suggests…" Unclear, please explain. Does 'only' refer to 'suggests' of the 1920-1930 warm period?

p. 1, l. 20: last sentence of abstract contradicts earlier statement of "..unprecedented melt…" after 1990.

p. 2, l. 5: Please also cite studies that imply a connection between subglacial meltwater injection and frontal ablation of marine terminating glaciers.

p. 2, l. 10: considered -> mentioned (?)

p. 2, l. 11: could not be -> is maybe not

p. 2, l. 19: attractive -> useful, powerful, robust (?)

p. 2, l. 23: validated -> evaluated (please use this throughout manuscript: models are by definition an approximation of reality and can therefore not be validated)

p. 3, l. 12: factional –> fractional

p. 4, l. 7: all of the. . ..I think it cannot be claimed that 'all' observations are really used. Perhaps the greatest fraction? Can this be supported by a reference?

p. 4, l. 12: surface marine winds -> near surface winds over the ocean surface

p. 4, l. 13: "As this reanalysis assimilates much less data than ERA-40/ERA-Interim," Is this also true for the overlapping period? Reliable -> accurate, reliability -> accuracy. Has the increase in accuracy been published in literature?

p. 4, l. 16: "covering the half of the last century" Second half.

p. 5, l. 3: Confusing: here summer T at 600 hPa is mentioned, in line 6 summer T at 700 hPa.

p. 5, l. 7: 'drives'. I suspect that what you mean to say is that T at 700 hPa is a good predictor for melt variability in MAR?

p. 5, l. 13: "Surprisingly, the comparison is worse with the 2nd generation of the NCEP reanalysis, which is warmer than ERA-Interim in summer except at the South-East of Greenland" I don't see this in Fig. 1f: the southeast is also too warm?

p. 5, l. 17: ". . .too warm (see Fig. 1c) and too cold (Fig. 1g). . ." I assume 'warm' and 'cold' must be swapped in this sentence.

p. 5, l. 21: Why is the performance of 20CRv2c not discussed here? It was not corrected? Using the acronyms '20CRv2-corr' and 'ERA-20C-corr' is somewhat confusing because the 'c' in '20CRv2c' presumably also stands for 'corrected'.

p. 5, l. 31: gauged -> represented

p. 6, l. 13: "Both ERA-20C forced simulations also significantly underestimate precipitation along the south-western coast. . ." This does not become clear from Fig. 4d.

p. 6, l. 19: "However, both simulations underestimate precipitation along the south-east coast with respect to MAR_ERA−Interim." But these deviations are also mostly

hatched, i.e. are not significant according to the definition used here?

p. 6, l. 26: the same -> similar

p. 6, l. 28: What does the '+40%' mean?

Caption Fig. 1 and elsewhere: Celsius degrees -> degrees Celcius

Caption Fig. 2: Please include explanation of the wind vectors in these plots, do they represent anomalies in the wind field?

---

## Author Comment (AC1) · 20 Jan 2017

We first would like to thank the reviewer comments which will help to improve our manuscript.

These results are certainly interesting and potentially important, but I have two main concerns. First, I found the paper very hard to read, mainly because of the writing style and the fact that the authors decided to show nearly all results rather than making an informed decision on the most relevant parameters and results.

We agree that a lot of results and statistics are presented. In function of the other reviewers comments, we could put in Supplementary Material the validation of the melt extent as well as several tables.

Second, the results are described in a merely qualitative fashion, without a more quantitative analysis of the reasons of the substantial differences between the various model runs. As a result a clear conclusion is lacking as to the quality of the re-analysis products for reproducing Greenland surface mass balance.

We think that the differences between the MAR simulations are well explained in respect to differences in the forcing reanalysis and the conclusions list the best reanalysis (pg 12, lines 14-15). However, we agree that additional general conclusions will be needed in the revised version of our manuscript. Eg: Which are the best reanalysis ? Over which periods the results are reliable ? … This is clearly mentioned in the revised conclusion of our manuscript.

These concerns are described in more detail below, together with some textual comments where clarification is needed (listing not complete). Major revisions will be necessary to bring the paper to a publishable level.

Major comments
The paper is very difficult to read, owing to the multitude of model acronyms and the overwhelming number of figure frames and tables contents. Already early in the paper I got confused by all MAR model versions presented. On page 3 alone, there is mention of MAR, MAR3.5.2, MARv2, MARv3.x, MARv3.2.

Between Fettweis et al. (2013) using MARv2 and this paper using MARv3.5.2, some MAR biases have been identified in several papers using intermediate MAR versions. Therefore, we judged important to list these biases (which have been corrected in MARv3.5.2) and to clearly mention the different MAR versions used in the text. Several papers used MAR outputs as comparison without giving the model version while each one has its own advantages and drawbacks. By explicitly listing here the MAR version used, we aim to inverse this trend.

. ... Some more sobering statistics: the manuscript contains six tables with about 500 numbers, sixty lines in line graphs, more than 40 maps, and the acronym MAR is used almost 300 times!

As proposed above, some comparisons can be added in the Supplementary Material. In addition, in function of the other reviewer remarks and editor suggestions, some less interesting MAR simulations (MAR forced by NCEP2, by JRA-55, by ERA-20C not corrected, by 20CRv2 not corrected) could be put in Supplementary Material although it is a pity to mask a part of these results showing sensitivity of the reanalysis used.

Readability can be further improved by not combining multiple results in a single sentence. Moreover, results should be easily traceable in figures. This now is not always the case. For example, p. 9, l. 17 reads: "MAR also underestimates accumulation in the south-east versus ice cores but overestimates versus BOX13 because this data set is based on RACMO2 outputs which are known to underestimate accumulation in this area (Noël et al., 2016)." This sentence starts by stating that MAR underestimates accumulation in the southeast compared to ice cores. But in Fig. 6c I see red colours, indicating an overestimation? The sentence continues that MAR overestimates (accumulation) vs. BOX 13, but Fig. 6c shows red and blue colours? Then it concludes that BOX13 is wrong in this area because RACMO2 outputs are underestimating accumulation in the southeast. So what is the conclusion?

Sorry, there is a mistake in the text. It is at the north-east and not at the south-east. In respect to Fig 6b (using observations only), most of the comparisons with ice cores in the accumulation zone are blue at the north-east while, over this same area, MAR overestimates BOX13 estimations (which is based on model results and not observations). Only ice core observations (shown with blue cross in Fig. 6a) are discussed here. The conclusion is that both MAR and BOX13 underestimates accumulation in this area. These sentences will be rephrased to be clearer in the revised version and the fact that only the accumulation observations are discussed here will be explicitly mentioned in the revised manuscript.

p. 3, l. 13: how can SMB be robustly calculated outside the ice sheet mask? If ice is assumed to be present in a certain grid cell that is currently tundra, then the local climate would be altered (cooled) and hence SMB would be influenced?

Indeed, there is a small influence but this impact can be neglected. For each MAR atmospheric pixel, there are 2 surface sub-pixels covered by tundra and permanent ice. At each time step, the surface energy balance and surface temperature are computed for each sub-pixel by the SISVAT surface model. They are averaged afterward over the whole atmospheric pixel for being used into MAR as input. Knowing that the average is weighted by respective cover of each sub-pixel and that the permanent ice cover over the "true" tundra is 0.001% (corresponding to the FORTRAN precision), we can resonantly assume that this impact is not significant in respect to a full tundra covered pixel.

This strategy allows to estimate SMB outside the MAR ice sheet mask (which is dependent of the spatial resolution) in the aim of forcing ice sheet models afterward. However, it is important to note that SMB for these "dominant" tundra pixels is underestimated because the near-surface temperature over these pixels is more representative of the tundra conditions than the ice sheet conditions and therefore, the melt is overestimated but it is better than nothing.

p. 3: Selecting the 700 hPa temperature as predictor for melt appears unfortunate, because this pressure level intersects with the ice sheet surface. This implies that T at 700 hPa at the lower parts of the ice sheet represents a free atmosphere temperature, at intermediate parts of the ice sheet it represents a boundary layer value (temperature inversion) and at the highest levels a below-surface (extrapolated?) value. This is confirmed by the blue lines in Fig. 1a, which show a conspicuous local minimum over the higher parts of the ice sheet, clearly caused by the ice sheet surface, a feature that would not be expected if a higher level (500 hPa) had been selected. The authors partly recognize this problem by masking out the level below 2000 m in Fig. 1, but that is again unfortunate because melt also does take place above this altitude so part of the interesting information is lost.

MAR is not directly sensitive to free atmosphere temperature biases over the Greenland ice sheet but rather at its lateral boundaries where the 850 hPa level is even more relevant as most of the melt occur below 1500m. But this level will be too masked by the ice sheet topography to be shown on a figure. The 500 hPa level is above the ice sheet summit but biases at this level less impact the melt amount simulated by MAR. Therefore, the 700 hPa is a good compromise to be shown although this level crosses the ice sheet topography. Anyway, the choice of the 600 or 700 hPa level to evaluate the melt variability is justified in Fettweis et al. (2013) and evaluations at other levels (850 and 500 hPa) than 700hPa will be provided in the Supplementary Material of the revised version.

[Figure]

On the figure above, you can see the mean difference over **1948**-**2010** of the JJA temperature at 850hPa (left), 700hPa (middle) and 500hPa (right) of 20CRv2 in respect to NCEP-NCARv1. We can see that 20CRv2 is too warm at each level.

p. 4: Why using the obsolete 20CRv2 at all when an improved/corrected product is available (20CRv2c)?

Using both 20CRv2 (where a temperature correction is needed) and 20CRV2c (where no temperature correction is applied) as forcing allows to show the sensibility (in particular at the beginning of the last century) of MAR results to very similar forcings as well as, in the same time, the uncertainties in the reanalysis. Knowing that there are only 2 available reanalysis before 1950, that the MAR results forced by 20CRv2 and 20CRv2c are significantly different before 1950, showing both time series is relevant for us. To improve the clarity of several figures, we could remove the JRA and NCEP2 forced time series as these simulations do not bring any new stuff in respect to the ERA and NCEP1 forced time series.

p. 5: A temperature correction of +/- 1 C is applied to 20CRv2 and ERA-20C, based on a 1980-1999 comparison with ERA-Interim. Can one be sure that these biases have been constant before and after

this period? Can you show how stable this bias has been in time for the full period 1958-2015 for which ERA-Interim and ERA-40 are available, which appear to perform adequately? Why is this only done for NCEPv1 (line 26)? Why are these results, which are instrumental for the interpretation of the results in this paper, not shown? The same question applies to the height of the 500 hPa level displayed in Fig. 2. How constant are these biases in time over the period with reliable forcing (∼1950-present)?

[Figure]

On the figure above, you can see the mean difference over  [90°W-0°W, 50°N-90°N] of the JJA temperature at 700hPa (in solid) and 500hPa (in dash) from 20CRv2 (in red) and from ERA-20C (in blue) in respect to NCEP-NCARv1 Reanalysis (1948-2010) (Unites are °C). As shown in Figs 1-2, NCEPv1 compares very well with ERA-Interim and ERA-40 (not shown here). We can see that the biases vary a few with time but 20CRv2 is every time too warm while ERA-20C is too cold. It is clear that applying a same correction over the whole period could induce additional biases because the quality of the reanalysis is not constant in time (we have no information to evaluate this one  before 1950). Nonetheless, a constant correction over time allows to compare different periods, knowing that we can assume that the corrected time series is homogeneous. This issue will be more discussed in depth in the manuscript and remember in the conclusion. Finally, it should be noted that no correction is applied when MAR is forced by 20CRv2c.

[Figure]

On the figure above, you can see the mean difference over **1948-2010** (left) and over **1980-1999** (right) of the annual mean geopotential height at 500 hPa from 20CRv2 (top) and ERA-20c (below) in respect to the NCEP-NCARv1 reanalysis (units are meters). As 20CRv2 (resp. ERA-20c) is too warm (resp. too cold), 20CRv2 (resp. ERA-20c) overestimates (resp. underestimates) the annual geopotential height at 500 hPa. However, these biases are constant in time and not impacted by the reference period chosen.

Similar figures will be added in the Supplementary Material of the revised version of our paper to valid our assumption.

p. 7: To improve the logical sequence of the paper, the evaluation of the ERA-Interim forced run (section 4) should be presented before the other results in Figs. 1-3.

An other solution should be to start by showing MAR results (Figs 3-4) and afterward explaining the differences between the MAR simulations in respect to discrepancies in reanalysis (Figs 1-2). But, we prefer to keep the logic used in Fettweis et al. (2013) where the forcings were discussed first. In addition, as MAR results using corrected reanalysis (20CRv2 and ERA20C) are shown in Fig. 3-4, keeping this sequence allows to justify afterward to show MAR results using corrected and not corrected reanalysis as forcing.

Figure 5: Knowing that absorbed shortwave radiation is the main energy source for melting, it is quite remarkable that ablation is so well reproduced in Fig. 5c, while surface albedo is clearly too high in MAR, by up to a factor of 2. This would imply that, all other things being equal, ablation would be significantly overestimated had albedo been correct. How can this be reconciled, are there compensating errors?

There are obviously compensating errors as it is the case in each climate model. MAR is firstly tuned to successfully simulate the SMB and ablation but not albedo. In this version (3.5.2) of MAR, MAR overestimates the bare ice albedo but underestimates longwave (LWD) and overestimates shortwave (SWD) because it underestimates the cloudiness (like RACMO for example). However MAR is able to successfully simulate ablation and (near) surface temperature, meaning that its Surface Energy Balance (SEB) is OK. But it is obvious that there are biases in the individual fluxes of SEB which, when they are summed, are compensated. In the next version of MAR, there are more clouds reducing the biases of SWD and LWD and allowing to have a lower bare ice albedo value better in agreement with that the observed one.

This problem of  compensating errors will be explicitly mentioned in the revised version of our paper in Section 4.1

Textual comments
p. 1, l. 9: validated -> support
ok, thanks. This will be corrected in the revised version of your paper.
p. 1, l. 13: The period 1961-1990 is not commonly chosen as reference period because there was approximate balance, rather it is the only official climatological period that can be chosen before significant changes occurred in Greenland.
ok
p. 1, l. 19: "stationarity assumption" Unclear, please explain.
We mean here that the SMB has been quite stable after 1930 until the end of the 1990's. This sentence will be rephrased.

p. 1, l. 20: ". . .only suggests. . ." Unclear, please explain. Does 'only' refer to 'suggests' of the 1920-1930 warm period?

We mean here that only the ERA-20C forced simulation suggest that … and not the 20CRv2(c) forced simulations. This sentence will be rephrased.

p. 1, l. 20: last sentence of abstract contradicts earlier statement of "..unprecedented melt. . ." after 1990.

SMB in the 1930's was "comparable" to SMB anomalies observed now but it was the results of BOTH positive anomaly of melt and negative anomalies of accumulation while currently, only melt anomalies drive the current SMB anomalies. Therefore, the current melt rates are well unprecedented.

p. 2, l. 5: Please also cite studies that imply a connection between subglacial meltwater injection and frontal ablation of marine terminating glaciers.

ok

p. 2, l. 10: considered -> mentioned (?)

ok

p. 2, l. 11: could not be -> is maybe not

ok

p. 2, l. 19: attractive -> useful, powerful, robust (?)

"powerful" is a better word indeed.

p. 2, l. 23: validated -> evaluated (please use this throughout manuscript: models are by definition an approximation of reality and can therefore not be validated)

ok

p. 3, l. 12: factional –> fractional

ok

p. 4, l. 7: all of the. . ..I think it cannot be claimed that 'all' observations are really used.
Perhaps the greatest fraction? Can this be supported by a reference?

greatest fraction is indeed more adequate here.

p. 4, l. 12: surface marine winds -> near surface winds over the ocean surface

ok

p. 4, l. 13: "As this reanalysis assimilates much less data than ERA-40/ERA-Interim," Is this also true for the overlapping period?

yes

Reliable -> accurate, reliability -> accuracy. Has the increase in accuracy been published in literature?

Yest, in the ERA_20C paper (Poli et al.,2016) which will be referenced here.

p. 4, l. 16: "covering the half of the last century" Second half.

ok

p. 5, l. 3: Confusing: here summer T at 600 hPa is mentioned, in line 6 summer T at 700 hPa.

ok

p. 5, l. 7: 'drives'. I suspect that what you mean to say is that T at 700 hPa is a good predictor for melt variability in MAR?

Yes. This sentence will be rephrased.

p. 5, l. 13: "Surprisingly, the comparison is worse with the 2nd generation of the NCEP reanalysis, which is warmer than ERA-Interim in summer except at the South-East of Greenland" I don't see this in Fig. 1f: the southeast is also too warm?

It is at the southeast of the domain presented here (Iceland). This sentence will be rephrased.

p. 5, l. 17: ". . .too warm (see Fig. 1c) and too cold (Fig. 1g). . ." I assume 'warm' and 'cold' must be swapped in this sentence.

The reference of the figure must be swapped indeed.

p. 5, l. 21: Why is the performance of 20CRv2c not discussed here? It was not corrected? Using the acronyms '20CRv2-corr' and 'ERA-20C-corr' is somewhat confusing because the 'c' in '20CRv2c' presumably also stands for 'corrected'.

The performance of 20CRv2c is discussed p5, line 28-20. We agree about he confusion 20CRv2c and 20CRv2-corr. The corrected reanalysis could be write CORR-ERA-20C and CORR-20CRv2.

p. 6, l. 13: "Both ERA-20C forced simulations also significantly underestimate precipitation along the south-western coast. . ." This does not become clear from Fig. 4d.

Latitude boundary will be added in the text to well specify that it the "south" part of the south-western coast.

p. 6, l. 19: "However, both simulations underestimate precipitation along the south-
east coast with respect to MAR_ERA−Interim." But these deviations are also mostly hatched, i.e. are not significant according to the definition used here?

These biases are not significant indeed.  This sentence will be rephrased.

p. 6, l. 26: the same -> similar

ok

p. 6, l. 28: What does the '+40%' mean?

40% means the runoff overestimate of MAR forced by 20CRv2 without correction. This sentence will be rephrased.

Caption Fig. 1 and elsewhere: Celsius degrees -> degrees Celcius

ok

Caption Fig. 2: Please include explanation of the wind vectors in these plots, do they
represent anomalies in the wind field?

They represent well anomalies of wind field. The caption will be updated.

---

## Referee Comment (RC2) · Anonymous Referee #2 · 4 Feb 2017

This is an original and thorough analysis of changes in Greenland Ice Sheet surface mass balance from 1900-2015 based on the regional climate model (RCM) MAR that was run with different climatic forcings from most currently-available reanalysis products (ECMWF ERA-Interim and ERA-40, 20CR, ERA-20C, NCEP/NCAR, JRA-55 etc.), with validation provided mainly based on PROMICE automatic weather station observations, and comparison of modelled melt with microwave satellite-derived melt extent.

Unsurprisingly there are significant differences in surface climate from the different RCM forcings but this kind of comparison is valuable as a current summary of the use and likely reliability of the various reanalysis products, as well as a useful guide for future work. The paper is therefore of significant interest to the GrIS community,

especially given the recent widespread adoption of MAR.

However, there are quite a number of (mainly minor) problems with the writing style that need to be corrected before publication, and in general the paper needs a thorough copy-edit. I list some corrections below. I do not see a need to move some of the scientific results (which are all interesting and best presented together) into "Supplementary Information".

page 1, lines 1/2: "decrease RELATIVE to last century" p.1, l.10 (& elsewhere): "data set" -> "dataset". p.1, l.11 : insert comma after "some biases remain in MAR". p.1, l.14 "SMB was anomalously positive (∼10%)" - I'm not sure it makes sense to have a percentage of SMB (which has no absolute zero reference point) - please clarify. p.1, l.17: "the result of an artefact in reanalysis THAT IS not WELL enough constrained". p.1, l.20: "Finally, ONLY the ERA-20C forced simulation suggests..."? p.2, l.6: should be "enhanced by Arctic amplification". p.2, l.11: insert comma after "since the end of the 1990's". p.2, l.14: "However, the NUMBER of in situ observations IS too sparse". p.2, l.28 "All PREVIOUS RCM-based SMB estimations". p.3, l.13 "ice sheet mask in MARv3.x allows THE COMPUTATION OF SMB outside the original MAR ice sheet mask (WITH the aim...". p.3, l.16 "weighted by the permanent cover of each grid cell (FOR CELLS covered by AT LEAST 50% of permanent ice)." - is this is what is meant? p.3, l.20: "with a minimum albedo SET to 0.7". p.3, l.24: "by slightly increasing the snowfall velocity, WHICH ENABLED more precipitation". p.4 list of reanalysis: was it also considered to use MERRA2 (state-of-the-art NASA reanalysis) in the comparisons for the 1979- period? p.5, l.10: "covered by all DATASETS used here and DURING WHICH SMB has been RELATIVELY stable". Add reference? NB: SMB was already starting to decline markedly during the late 1990s. p.5, l.17: I think that fig. 1c and fig. 1f references are the wrong way round here - please check. p.5, l.28: do you mean "enable a better comparison of MAR with in situ temperature measurements THAN WITH using unmodified 20CRv2 and ERA-20C..."? p.5, l.34: add comma after "which underestimates wind speed at 500 hPa". p.6, l.6: "However, when looking AT spatial differences". p.6, l.8: "MAR_ERA-40 SLIGHTLY OVERESTIMATES precipitation". p.6, l.9 "ERA-40 humidity scheme, WHICH WERE LATER corrected in ERA-Interim". p.6, l.24: "NCEPv2 relative humidity and IS then affected". p.6, l.32: "overestimates" -> "overestimate". p.7, l.8: "12 AWS's listed in Table 2 THAT HAVE an elevation difference WITHIN 100m OF the interpolated MAR". p.7, l.10 "ON average FOR the 12 AWS's's". p.7, l.14: "MAR SLIGHTLY OVERESTIMATES". p.7, l.18: "(A bias of -18W/mˆ2, COM-PARED WITH A daily variability OF 43 W/mˆ2)". p.7, l.24 "Using other reanalyses (APART FROM ERA-INTERIM) as MAR forcing". p.7, l.26: "compare the best" -> "show the best agreement with PROMICE". p.7, l.27 "compares the worst" -> "shows the worst". p.8, l.5: "corrected AS A function of the elevation difference". p.8, l.9: "The data are not converted TO m W.E./yr". p.8, l.10: "with an elevation difference with the MAR topography OF LESS than 500m". p.8, l.16: "IN CONTRAST to the MAR-based reconstructions". p.8, l.29: "instead 0.4-0.55 INDICATE LATER CORRECTION of this overestimation". p.9, l.12: "recent decades, although the AMOUNT of assimilated data is larger. The LOWEST correlations...". p.9, l.18: "but overestimates versus BOX13 because THE LATTER dataset". p.9, l.25: "underestimates accumulation RELATIVE to BOX13 (WHERE THE LATTER IS based on RACMO2)." p.9, l.29: "suggesting that FURTHER accumulation measurement campaigns". p.10, l.18: delete "rather". p.10, l.19: "assimilated into BOX13, THE LATTER reconstruction perfectly matches". p.10, l.23: two cases of "MAR_20CRv2c" are mentioned but should these both be the same (is one of them possibly a typo)? p.10, l.26: "at end of the 1970's, WHICH ARE overestimated by...". p.11, l.20: "but IS LESS PRONOUNCED than in the MAR simu-lations". p.11, l.22: "suggesting that THE lower the amount of assimilated data, THE higher the spread". p.11, l.30: "part of this increase could just be due to AN artefact in THE 20CRv2(c)." p.12, l.15 "while NCEP-NCARv1 outperforms ERA-40/ERA-Interim since the 1950s" - do you mean more specifically from 1950 to 1980? p.12, l.19: "the highest SMB rates are reached over the 1970s-EARLY 1990s". p.12, l.25: "SIMILAR discrepancies can be seen in the MAR simulated". p.12, ll.27/28: "but TO a lessER extent than MAR, while Hanna et al. (2011)...". p.13, l.6: "the amount of DATA assimilated into ERA-20C". p.13, l.7 "without enough gauge observations" - how many is "enough"? p.13, l.15: "suggesting that mass gain MAY WELL HAVE OCCURRED during this period, in agreement with...". p.13, l.18: "IS unprecedented". Figure 1 caption, change last sentence to: "DUE TO the aim of ONLY showing comparisons in the free atmosphere (700 hPa), and the datasets...". Table 1 caption, last two sentences correct to "The RUNOFF is the FRACTION of water from both surface melt and rainfall THAT IS NOT REFROZEN BEFORE reaching the ocean...asterisk WAS corrected..."

---

## Author Comment (AC2) · 9 Feb 2017

We first would like to thank the reviewer for his very usefull comments which will help a lot to improve our manuscript.

This is an original and thorough analysis of changes in Greenland Ice Sheet surface mass balance from 1900-2015 based on the regional climate model (RCM) MAR that was run with different climatic forcings from most currently-available reanalysis products (ECMWF ERA-Interim and ERA-40, 20CR, ERA-20C, NCEP/NCAR, JRA-55 etc.), with validation provided mainly based on PROMICE automatic weather station observations, and comparison of modelled melt with microwave satellite-derived melt extent. Unsurprisingly there are significant differences in surface climate from the different RCM forcings but this kind of comparison is valuable as a current summary of the use and likely reliability of the various reanalysis products, as well as a useful guide for future work. The paper is therefore of significant interest to the GrIS community, especially given the recent widespread adoption of MAR.

Thanks.

However, there are quite a number of (mainly minor) problems with the writing style that need to be corrected before publication, and in general the paper needs a thorough copy-edit. I list some corrections below.

Thanks. We will pay a special attention to copy-editing in the revised version of our paper.

I do not see a need to move some of the scientific results (which are all interesting and best presented together) into "Supplementary Information".

Thanks. If the editor agrees also, we will leave the current presented materials as they are.

page 1, lines 1/2: "decrease RELATIVE to last century"
p.1, l.10 (& elsewhere): "data set" -> "dataset".
p.1, l.11 : insert comma after "some biases remain in MAR".

OK, thanks.

p.1, l.14 "SMB was anomalously positive (∼10%)" - I'm not sure it makes sense to have a percentage of SMB (which has no absolute zero reference point) - please clarify.

Indeed, we will list explicitly the SMB anomalies in GT/yr as well as the SMB used as reference.

p.1, l.17: "the result of an artefact in reanalysis THAT IS not WELL enough constrained".
p.1, l.20: "Finally, ONLY the ERA-20C forced simulation suggests..."?
p.2, l.6: should be "enhanced by Arctic amplification".
p.2, l.11: insert comma after "since the end of the 1990's".
p.2, l.14: "However, the NUMBER of in situ observations IS too sparse".
p.2, l.28 "All PREVIOUS RCM-based SMB estimations".
p.3, l.13 "ice sheet mask in MARv3.x allows THE COMPUTATION OF SMB outside the original MAR ice sheet mask (WITH the aim...".
OK, thanks.

p.3, l.16 "weighted by the permanent cover of each grid cell (FOR CELLS covered by AT LEAST 50% of permanent ice)." - is this is what is meant?

Yes, thanks

p.3, l.20: "with a minimum albedo SET to 0.7".
p.3, l.24: "by slightly increasing the snowfall velocity, WHICH ENABLED more precipitation".

Ok, thanks.

p.4 list of reanalysis: was it also considered to use MERRA2 (state-of-the-art NASA reanalysis) in the comparisons for the 1979- period?

MERRA2 was not available when these MAR simulations were performed. First tests of MARv3.6 forced by ERA-Interim vs MAR forced by MERRA2 over Antarctica suggest not significant differences over 1979-2015. Therefore, adding MERRA2 as forcing in our paper will not change the story of this paper knowing that only the 1979-2016 period (when all reanalyses agree) is covered by this reanalysis. However, when ERA5 will be available (in a couple of months we hope), we plan to test MAR at higher resolutions (~ 10km) to test the interest of these new higher resolution reanalyses than ERA-Interim as forcing.

p.5, l.10: "covered by all DATASETS used here and DURING WHICH SMB has been RELATIVELY stable". Add reference? NB: SMB was already starting to decline markedly during the late 1990s.

We fully agree that SMB was already started to decrease at the end of 1980-1999 which has been also used as reference period in Fettweis et al. (2013). This reference will be added in the text. The period 1980-1999 is in fact the "less worst" reference period. 1960-1990 can not be used because it is not covered by the recent reanalyses (ERA-Int and NCEP2). The 2000's years should be avoided in a reference period over GrIS. Finally, having a reference period shorter than 20 yrs can not be justified in climatology. However, as shown in Fettweis et al. (2013), changing the reference period will not impact the story of this paper.

p.5, l.17: I think that fig. 1c and fig. 1f references are the wrong way round here - please check.

Indeed, the references to figures should be inverted. Thanks.

p.5, l.28: do you mean "enable a better comparison of MAR with in situ temperature measurements THAN WITH using unmodified 20CRv2 and ERA-20C..."?

Yes. We will reformulate this sentence.

p.5, l.34: add comma after "which underestimates wind speed at 500 hPa".
p.6, l.6: "However, when looking AT spatial differences".
p.6, l.8: "MAR_ERA-40 SLIGHTLY OVERESTIMATES precipitation".
p.6, l.9 "ERA-40 humidity scheme, WHICH WERE LATER corrected in ERA-Interim".

p.6, l.24: "NCEPv2 relative humidity and IS then affected".
p.6, l.32: "overestimates" -> "overestimate".
p.7, l.8: "12 AWS's listed in Table 2 THAT HAVE an elevation difference WITHIN 100m OF the interpolated MAR".
p.7, l.10 "ON average FOR the 12 AWS's".
p.7, l.14: "MAR SLIGHTLY OVERESTIMATES".
p.7, l.18: "(A bias of -18W/m^2, COMPARED WITH A daily variability OF 43 W/m^2)".
p.7, l.24 "Using other reanalyses (APART FROM ERA-INTERIM) as MAR forcing".
p.7, l.26: "compare the best" -> "show the best agreement with PROMICE".
p.7, l.27 "compares the worst" -> "shows the worst".
p.8, l.5: "corrected AS A function of the elevation difference".
p.8, l.9: "The data are not converted TO m W.E./yr".
p.8, l.10: "with an elevation difference with the MAR topography OF LESS than 500m".
p.8, l.16: "IN CONTRAST to the MAR-based reconstructions".
p.8, l.29: "instead 0.4-0.55 INDICATE LATER CORRECTION of this overestimation".
p.9, l.12: "recent decades, although the AMOUNT of assimilated data is larger. The LOWEST correlations...".
p.9, l.18: "but overestimates versus BOX13 because THE LATTER dataset".
p.9, l.25: "underestimates accumulation RELATIVE to BOX13 (WHERE THE LATTER IS based on RACMO2)."
p.9, l.29: "suggesting that FURTHER accumulation measurement campaigns".
p.10, l.18: delete "rather".
p.10, l.19: "assimilated into BOX13, THE LATTER reconstruction perfectly matches".

Ok, thanks. All of these ones will be corrected.

p.10, l.23: two cases of "MAR_20CRv2c" are mentioned but should these both be the same (is one of them possibly a typo?)?

Yes. The second case is MAR_20CRv2-corr. We will double check in the revised manuscript MAR_20CRv2-corr vs  MAR_20CRv2c. As highlighted by Reviewer #1, v2c vs v2-corr is confusing. Perhaps -CORR vs -corr will be more clear.

p.10, l.26: "at end of the 1970's, WHICH ARE overestimated by...".
p.11, l.20: "but IS LESS PRONOUNCED than in the MAR simulations".
p.11, l.22: "suggesting that THE lower the amount of assimilated data, THE higher the spread".
p.11, l.30: "part of this increase could just be due to AN artefact in THE 20CRv2(c)."
p.12, l.15 "while NCEP-NCARv1 outperforms ERA-40/ERA-Interim since the 1950s" - do you mean more specifically from 1950 to 1980?

yes. The problem is indeed ERA-40 in the ECMWF forced MAR time series. This sentence will be rephrased.

p.12, l.19: "the highest SMB rates are reached over the 1970s-EARLY 1990s".
p.12, l.25: "SIMILAR discrepancies can be seen in the MAR simulated".
p.12, ll.27/28: "but TO a lessER extent than MAR, while Hanna et al. (2011)...".
p.13, l.6: "the amount of DATA assimilated into ERA-20C".

p.13, l.7 "without enough gauge observations" - how many is "enough"?

Good question!! We have no idea. The problem is mainly that there is no observation in the areas where the differences are the highest. Having 2-3 obs in these areas are likely enough if the obs agree between them. We will rephrase this sentence to be less vague.

p.13, l.15: "suggesting that mass gain MAY WELL HAVE OCCURRED during this period, in agreement with...".
p.13, l.18: "IS unprecedented". Figure 1 caption, change last sentence to: "DUE TO the aim of ONLY showing comparisons in the free atmosphere (700 hPa), and the datasets...". Table 1 caption, last two sentences correct to "The RUNOFF is the FRACTION of water from both surface melt and rainfall THAT IS NOT REFROZEN BEFORE reaching the ocean...asterisk WAS corrected…"

Ok, thanks.

---

## Author Response (AR2)

Dear Editor, Dear Edward,

You will find a new version of our manuscript including the last corrections from Reviewer #1 as well as an additional copy-editing more in depth. A version showing changes in the text is included in this PDF.

All the best,

Xavier on behalf of the coauthors

[revised manuscript text omitted]

---

## Author Response (AR3)

Dear Editor, Dear Edward,

You will find a new version of our manuscript including your corrections as well as a rephrasing of the litigious sentence:

p.10, l.29: "MAR also underestimates accumulation compared to ice cores in the north-east but is better than the BOX13 results, which significantly ** overestimate** the accumulation because they are based on RACMO2 outputs, which are known to underestimate accumulation in this area (Noël et al., 2016)."

Sorry, there was a big mistake in this sentence. Both models underestimate accumulation in this area ant not MAR underestimates and BOX13 overestimates.

[Figure]

b) SMB biases from $MAR_{NCEPv1}$          c) SMB: $MAR_{NCEPv1}$ − BOX13 d

Therefore, we have reformulate this sentence by:

[revised manuscript text omitted]